# Thiazoline-related innate fear stimuli orchestrate hypothermia and anti-hypoxia via sensory TRPA1 activation

Tomohiko Matsuo[1], Tomoko Isosaka[1], Yuichiro Hayashi[1], Lijun Tang[1], Akihiro Doi [1], Aiko Yasuda[1], Mikio Hayashi [2], Chia-Ying Lee[3], Liqin Cao[3], Natsumaro Kutsuna[4,5], Sachihiro Matsunaga [6], Takeshi Matsuda[7], Ikuko Yao[7], Mitsuyoshi Setou[8], Dai Kanagawa[1], Koichiro Higasa[9], Masahito Ikawa [10], Qinghua Liu [3,11✉], Reiko Kobayakawa [1✉] & Ko Kobayakawa [1✉]

Thiazoline-related innate fear-eliciting compounds (tFOs) orchestrate hypothermia, hypometabolism, and anti-hypoxia, which enable survival in lethal hypoxic conditions. Here, we show that most of these effects are severely attenuated in transient receptor potential ankyrin 1 (*Trpa1*) knockout mice. TFO-induced hypothermia involves the *Trpa1*-mediated trigeminal/vagal pathways and non-*Trpa1* olfactory pathway. TFOs activate *Trpa1*-positive sensory pathways projecting from trigeminal and vagal ganglia to the spinal trigeminal nucleus (Sp5) and nucleus of the solitary tract (NTS), and their artificial activation induces hypothermia. TFO presentation activates the NTS-Parabrachial nucleus pathway to induce hypothermia and hypometabolism; this activation was suppressed in *Trpa1* knockout mice. TRPA1 activation is insufficient to trigger tFO-mediated anti-hypoxic effects; Sp5/NTS activation is also necessary. Accordingly, we find a novel molecule that enables mice to survive in a lethal hypoxic condition ten times longer than known tFOs. Combinations of appropriate tFOs and TRPA1 command intrinsic physiological responses relevant to survival fate.

[1] Department of Functional Neuroscience, Institute of Biomedical Science, Kansai Medical University, Osaka, Japan. [2] Department of Cellular and Functional Biology, Institute of Biomedical Science, Kansai Medical University, Osaka, Japan. [3] International Institute for Integrative Sleep Medicine (WPI-IIIS), University of Tsukuba, Tsukuba, Ibaraki, Japan. [4] Department of Integrated Biosciences, Graduate School of Frontier Sciences, University of Tokyo, Chiba, Japan. [5] LPixel Inc., Tokyo, Japan. [6] Department of Applied Biological Science, Faculty of Science and Technology, Tokyo University of Science, Chiba, Japan. [7] Department of Optical Imaging, Institute for Medical Photonics Research, PMPERC and IMIC, Hamamatsu University School of Medicine, Shizuoka, Japan. [8] Department of Cellular and Molecular Anatomy and IMIC, Hamamatsu University School of Medicine, Shizuoka, Japan. [9] Department of Genome Analysis, Institute of Biomedical Science, Kansai Medical University, Osaka, Japan. [10] Research Institute for Microbial Diseases, Osaka University, Osaka, Japan. [11] National Institute of Biological Sciences, Beijing, China. ✉email: linqinghua@nibs.ac.cn; kobayakr@hirakata.kmu.ac.jp; kobayakk@hirakata.kmu.ac.jp

I n life-threatening situations, organisms protect themselves using latent bioprotective capabilities. The diving reflex, induced by trigeminal activation by cold immersion in near-drowning, and the vagal reflex, induced by fear perception, cause considerable fluctuations in the homeostatic state in humans[1,2]. These reflexes have supposedly evolved as life-protective abilities; however, clinical applications utilizing these effects have yet to be established.

Innate fear is intimately connected to the preservation of life. However, the absence of its effective inducers prevents uncovering latent protective abilities. By optimizing the chemical structure of 2,4,5-trimethyl-3-thiazoline (TMT)[3], a predator-derived compound and innate fear inducer in rodents, we developed artificial compounds, thiazoline-related fear-eliciting compounds (tFOs), e.g., 2-methyl-2-thiazoline (2MT), inducing potent innate fear response in mice[4]. We recently found that tFOs orchestrate robust hypothermia, anaerobic metabolism, and anti-hypoxic responses, which extended survival in lethal hypoxic conditions and decreased the severity of ischemia/reperfusion models[5]. Thus, identifying the receptor gene responsible for these physiological responses induced by tFOs is important to evaluate whether the phenomena can be used for medical applications.

Previous studies have identified several candidate receptor genes and neural pathways in the perception of tFOs. The dorsal olfactory pathway and its odorant receptors regulate avoidance and fear-related behaviors induced by TMT and the alarm pheromone 2-sec-butyl-2-thiazoline (SBT)[6–8]. On the other hand, forward genetic screenings identified that freezing and avoidance behaviors induced by tFOs, such as 2MT and TMT, were regulated by the transient receptor potential ankyrin 1 (Trpa1) gene in trigeminal neurons[9]. Furthermore, fear-related behaviors were also suppressed in Trpa1$^{-/-}$ mice in response to a natural product, e.g., snake-derived compounds. Thus, fear-related behaviors in response to tFOs are thought to be regulated by at least two different systems: (1) Trpa1 in the trigeminal neurons and (2) odorant receptors in the main olfactory system. However, the genes and neural pathways responsible for tFO-induced physiological effects are unknown. Here, we aimed to clarify the contribution of Trpa1 in regulating these effects.

TRPA1 was initially identified as a cold-activated ion channel[10]. TRPA1 is also activated by extrinsic stimuli, such as allyl isothiocyanate (AITC), a pungent component of mustard oil and wasabi, and formalin, a nociceptive stimulus[11,12], as well as by intrinsic stimuli, such as 4-hydroxy-2-nonenal and $H_2O_2$ generated by inflammation[13,14]. Furthermore, Trpa1 is involved in inflammatory pain and hypersensitivity after inflammation[15–17], in the perception of aberrant oxygen concentration, and in the regulation of respiratory responses to mild hypoxia[18,19]. Collectively, Trpa1 is considered an alarm sensor detecting multiple signals to transduce pain or danger information to the brain[18,20,21]. Extending these findings, we hypothesized that Trpa1 also has a crucial role in tFO-mediated latent physiological responses, which increase survival in lethal conditions. We found that tFOs are perceived by Trpa1 in the trigeminal and vagus nerves. This information is transmitted to the spinal trigeminal tract (Sp5) and the nucleus of the solitary tract (NTS) to regulate hypothermia, hypoxic metabolism, and survivability in lethal hypoxic conditions. Finally, by monitoring the activation of TRPA1 and Sp5/NTS, we identified a novel compound that could prolong survival in hypoxic conditions ten times longer than known tFOs. Taken together, our results indicate that Trpa1 not only functions as a danger sensor, but also commands the induction of physiological responses relevant to innate fear, and is even involved in the acquisition of viability in lethal hypoxic conditions.

## Results

### tFO induces hypothermia/anti-hypoxia via Trpa1.
Avoidance and risk assessment behaviors relevant to innate fear induced by 2MT are regulated by Trpa1 (refs. [4,9]). 2MT also induces robust physiological responses such as hypothermia and bradycardia[5]. We examined whether these physiological responses are regulated by Trpa1 using Trpa1$^{-/-}$ mice. Whereas body temperature in the baseline condition was not significantly altered, 2MT-induced hypothermia was greatly suppressed in Trpa1$^{-/-}$ mice (Fig. 1a, b). Most Trpa1-expressing cells co-express Trpv1 (ref. [10]). However, 2MT-induced hypothermia was not affected in Trpv1$^{-/-}$ mice (Fig. 1c). Hypothermia is observed not only because of tFO stimulation but also in response to restraint in a tight space, another type of innate fear stimulus[5]. If Trpa1 works as a peripheral sensor for 2MT to induce hypothermia, it is expected that hypothermia induced by other types of innate fear stimuli would not be affected in Trpa1$^{-/-}$ mice. As expected, restraint in a tight place induced hypothermia in Trpa1$^{-/-}$ mice (Fig. 1d). Similarly, the heart rate in the control condition was not altered, but quick and robust bradycardia following 2MT presentation was almost completely suppressed in Trpa1$^{-/-}$ mice (Fig. 1e). In contrast, reduction of heart rate induced by restraint in a tight space was observed in Trpa1$^{-/-}$ mice (Fig. 1f). Interestingly, hypothermia and bradycardia induced by restraint in a tight space tended to be greater in Trpa1$^{-/-}$ mice than those in control mice. Thus, it is suggested that the Trpa1 gene contributes to the initiation of innate fear-relevant suppression of body temperature/heart rate; Trpa1 may also have important roles in maintaining appropriate suppression levels to various stimuli.

Under normal conditions, mice cannot survive in a hypoxic environment, whereas it is reported that hydrogen sulfide ($H_2S$) prolongs the survival time in such environment[22]. High concentrations of $H_2S$ lead to the death of individuals by inhibiting the activity of the electron transfer chain in the mitochondria[23], which is essential for oxygen respiration. Low concentration of $H_2S$ is thought to partially inhibit mitochondrial respiratory activity, thereby enhancing the ability to survive in a hypoxic environment[22,24]. On the other hand, 2MT has no inhibitory effects on the mitochondrial respiratory chain. Nevertheless, 2MT-stimulation evokes hypoxic metabolism and hypoxia resistance ability[5]. How does 2MT induce hypoxia resistance without inhibiting the mitochondrial respiratory chain? As we reported previously, oxygen consumption was suppressed by presentation of 2MT. Importantly, prior presentation of 2MT increased the survival time in lethal 4% oxygen conditions[5]. Then, we asked if these impacts of 2MT presentation on oxygen metabolism are also controlled by Trpa1 gene. In Trpa1$^{-/-}$ mice, suppression of oxygen consumption induced by 2MT presentation was completely suppressed (Fig. 2a). Further, survival time in 4% oxygen condition was significantly shortened in 2MT-Trpa1$^{-/-}$ mice (Fig. 2b). These results suggest that 2MT causes hypometabolism to fulfill anti-hypoxic effects through Trpa1. These results also raise the possibility Trpa1 agonists other than tFOs might induce anti-hypoxic effects. To test this possibility, we analyzed the effects of previously reported Trpa1 agonists: trans-$\Delta^9$-tetrahydrocannabinol, a component of cannabis; AITC; and acetaminophen, an analgesic antipyretic[11,25,26]. Survival time under hypoxic conditions was prolonged by these ligands (Fig. 2c), and acetaminophen induced the strongest effect, which was absent in Trpa1$^{-/-}$ mice (Fig. 2d). Collectively, TRPA1 activation by diverse ligands orchestrate hypothermia, hypometabolism, and anti-hypoxic effects.

### Multiple sensory pathways are involved in tFO-induced hypothermia.
Both the main olfactory neurons and the trigeminal neurons regulate fearful behaviors evoked by

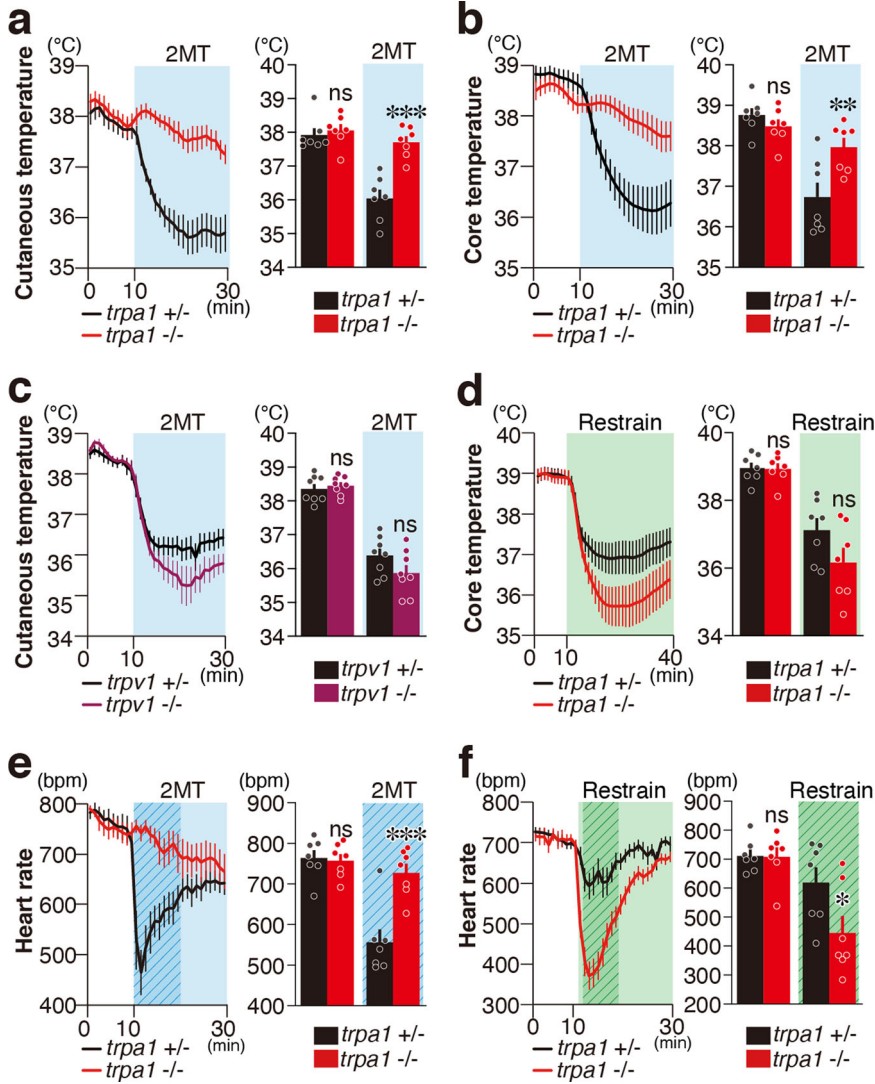

**Fig. 1 _Trpa1_ mediates tFO-evoked hypothermia and bradycardia. a, b** Cutaneous temperature (**a**) and core body temperature (**b**) temporal analysis in $Trpa1^{-/-}$ (red) and control (black) mice in response to presentation of 2-methyl-2-thiazoline (2MT) (**a**, $n = 7$ for each genotype; **b**, $n = 7$ for each genotype). Mean cutaneous/core temperature in 10 min of baseline session and 20 min of 2MT presentation are also shown (**a**, $p = 0.8708$ for baseline session and $p < 0.0001$ for during 2MT presentation; **b**, $p = 0.6638$ for baseline session and $p = 0.0026$ for 2MT session). **c** Cutaneous temperature temporal analysis in $Trpv1^{-/-}$ (purple) and control (black) mice in response to 2MT presentation ($n = 8$ for each genotype). Mean cutaneous temperature in 10 min of baseline session and 20 min of 2MT presentation are also shown ($p = 0.9121$ for baseline session and $p = 0.0935$ for 2MT session). **d** Core body temperature temporal analysis in $Trpa1^{-/-}$ (red) and control (black) mice in response to the restrained condition ($n = 7$ for each genotype). Mean core temperature during baseline session (1–10 min, $p = 0.9981$) and during restrained condition (12–40 min, $p = 0.0631$) are also shown. **e, f** Temporal analysis of heart rates in $Trpa1^{-/-}$ (red) and control (black) mice in response to 2MT presentation (**e**, $n = 7$ for each) and in restrained condition (**f**, $n = 7$ for each). Mean heart rate in response to 2MT presentation [**e**, $p = 0.9757$ for baseline session (1–10 min) and $p < 0.0001$ for 2MT session (11–20 min, marked by shaded duration in the left figure)] and the restrained condition [**f**, $p = 0.9987$ for baseline (1–10 min) and $p = 0.01084$ for restrained condition (12–20 min, marked by shaded areas in the left figure)] are also shown. Data are shown as mean ± SEM. Two-way ANOVA followed by Sidak's multiple comparison test was used to assess significance; *$p < 0.05$; **$p < 0.01$; ***$p < 0.001$; ns $p > 0.05$.

predator-derived compounds and tFOs[6–8]. However, the contribution of these neurons in regulating tFO-induced physiological responses is unclear. In olfactory bulbectomized (OBx) mice, cutaneous temperature reduction in response to 2MT exposure was suppressed compared to controls (Fig. 3a and Supplementary Fig. 1). The surgical removal of the olfactory bulbs (OBs) might injure parts of the trigeminal nerve innervating into the nasal cavity. To exclude this possibility, we analyzed the ΔD mutant mice, in which the dorsal zone olfactory sensory neurons (OSNs) that regulate avoidance behaviors to predator odorants are genetically depleted[6]. We observed suppressed cutaneous temperature reduction in response to 2MT presentation (Fig. 3b).

The cyclic nucleotide-gated olfactory (_Cnga2_) channel is critical for OSNs to generate odor-induced action potentials[27]. We also analyzed the ΔD(cng) mice, in which _Cnga2_ channel is deleted in the dorsal OSN[7]. Cutaneous temperature reduction was suppressed in these animals (Supplementary Fig. 2). Next, we analyzed the contribution of the trigeminal system in regulating cutaneous temperature. Mice receiving bilateral trigeminal ganglion (TG) electrocauterization died several days after surgery. In mice that received unilateral TG electrocauterization (ulTGx), we observed suppressed reduction of cutaneous temperature in response to 2MT, although contralateral TG and OBs are intact in these animals (Fig. 3c). Mice that received bilateral vagal nerve

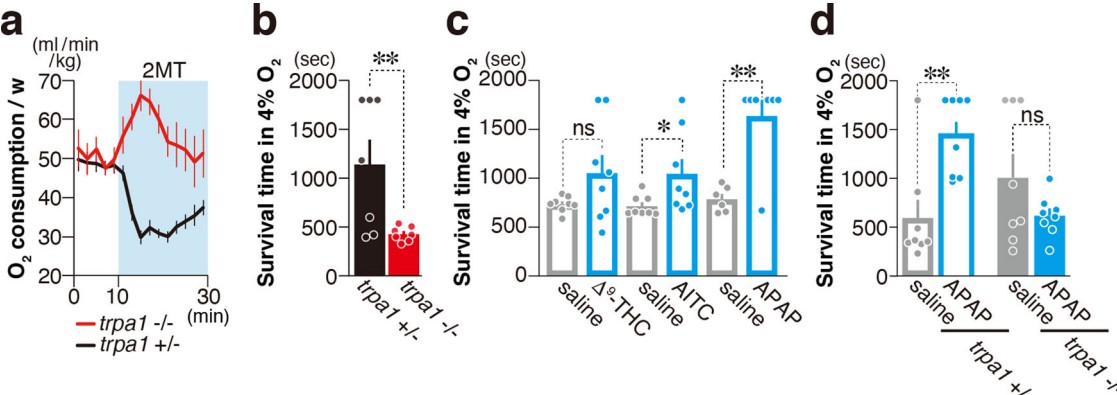

**Fig. 2 *Trpa1* mediates tFO-evoked anti-hypoxia. a** Temporal analysis of oxygen consumption in *Trpa1*$^{-/-}$ (red) and control (black) mice in response to 2MT presentation (**a**, $n = 9$ for *Trpa1*$^{+/-}$ and $n = 7$ for *Trpa1*$^{-/-}$). 2MT was presented at 10 min. **b** Mean survival time of *Trpa1*$^{-/-}$ (red) and control (black) mice in 4% oxygen with 10 min of prior 2MT presentation ($n = 7$ for each genotype, $p = 0.0122$). **c** Mean survival time of wildtype mice in 4% oxygen with (blue) and without (gray) intraperitoneal (IP) administration of the indicated compounds [$\Delta^9$-tetrahydrocannabinol (THC); $n = 8$ for each, $p = 0.0586$: isothiocyanate (AITC); $n = 8$ each, $p = 0.0258$: acetaminophen (APAP); $n = 6$ for saline and $n = 7$ for APAP, $p = 0.003$]. **d** Mean survival time of *Trpa1*$^{-/-}$ (closed bars; $n = 8$ each) and control (open bars; $n = 8$ each) mice in 4% oxygen with (blue) and without (gray) IP administration of APAP; $p = 0.0066$ for Trpa1$^{+/-}$ and $p = 0.2263$ for *Trpa1*$^{-/-}$. Data are shown as mean ± SEM. Log-rank test was used to assess statistical significance; *$p < 0.05$, **$p < 0.01$, ***$p < 0.001$.

transection at the cervical level also died immediately after surgery. In contrast to ulTGx, reduction of cutaneous temperature in response to 2MT presentation was not affected in mice receiving unilateral transection (Supplementary Fig. 3a). Then, we ablated the bilateral subdiaphragmatic vagus nerves (VNSs), and again observed suppression of 2MT-induced cutaneous temperature reduction (Supplementary Fig. 3b). Taken together, these results indicate that the main olfactory, trigeminal, and vagus systems are involved in tFO-induced hypothermia.

*Trpa1* is expressed not only in trigeminal and vagus nerves, but also in the olfactory epithelium[11,28,29]. Therefore, we subsequently addressed which of these systems are involved in the regulation of *Trpa1*-dependent tFO-induced hypothermia. Reduction of cutaneous temperature in response to 2MT was not affected in conditional knockout mice in which *Trpa1* was removed in OSNs by using olfactory marker protein (OMP)-Cre mice[30] (Fig. 3d). However, this temperature reduction was suppressed in mutant mice in which *Trpa1* was removed in the peripheral sensory neurons other than OSNs using Advillin-Cre mice[31] (Fig. 3e). In trigeminal neurons, most *Trpa1*$^+$ neurons coexpress *Trpv1* (ref. [11]). The administration of resiniferatoxin (RTX), an ultrapotent TRPV1 agonist, leads to cytotoxic calcium overload and cell death of *Trpv1*$^+$ neurons[32,33]. RTX administration in the TG suppressed 2MT-induced hypothermia (Fig. 3f). These results indicate that multiple sensory inputs, including the olfactory, trigeminal, and vagus nerves, are involved in tFO-induced hypothermia.

**Vagal and trigeminal afferent projections of *Trpa1*$^+$ neurons.** Hypothermia is regulated by the median preoptic nucleus (MnPO) and ventromedial preoptic area (VMPO) neurons in the hypothalamus[34–37]; hence, 2MT-induced hypothermia may similarly be regulated by these brain regions. To test this possibility, we analyzed the *c-fos* mRNA expression in control and *Trpa1*$^{-/-}$ mice in response to 2MT presentation (Supplementary Fig. 4a–c). 2MT stimulation upregulated *c-fos* mRNA expression in the VMPO. However, this increased expression was unaffected in *Trpa1*$^{-/-}$ mice, suggesting that 2MT-induced hypothermia mediated by *Trpa1* is regulated by brain regions other than the MnPO and VMPO in the hypothalamus. Recently, we showed that 2MT activates the neuronal pathway from the NTS in the

brainstem to the parabrachial nucleus (PBN) in the midbrain, and that artificial activation of this pathway induced hypothermia[5]. Thus, in this study, we considered that 2MT-induced activation of the NTS–PBN pathway may be regulated by *Trpa1* (Supplementary Fig. 4d–g). We found that 2MT-induced *c-fos* mRNA expression in the PBN and NTS was almost completely abolished in *Trpa1*$^{-/-}$ mice, indicating that 2MT-induced activation of the NTS–PBN pathway is mediated by *Trpa1*. These results suggest that 2MT-induced hypothermia mediated by *Trpa1* is regulated by the brainstem–midbrain pathway rather than the known hypothalamic thermoregulatory center. Thus, we aimed to elucidate the signaling pathway of *Trpa1*$^+$ neurons to the brainstem.

2MT presentation induced expression of *c-fos*, a neural activity marker, in both Sp5 and NTS[5], areas which receive axonal projections mainly from the trigeminal[38,39] and vagus nerves[40–42]. *Trpa1* is expressed in a subset of trigeminal and vagus nerve neurons[11,28], whose projection sites are not fully characterized. To address this question, we generated knock-in mice, in which the coding sequence of the *Trpa1* gene was replaced with that of the *Cre* gene. To visualize *Trpa1*-expressing cells, we crossed *Trpa1-Cre* mice with a reporter mouse strain expressing enhanced yellow fluorescent protein (EYFP) in a Cre-dependent manner (Fig. 4a). The cell bodies of EYFP$^+$ neurons were observed in both the trigeminal and vagus ganglia (TG and VG) by whole-mount imaging (Fig. 4b$_1$, b$_2$, c$_1$, c$_2$). Histochemical analysis of TG/VG sections confirmed EYFP expression in small-sized cells as reported previously[10,28] (Fig. 4b$_3$, c$_3$). Next, we analyzed the axonal projection of *Trpa1*$^+$ neurons in the Sp5. EYFP$^+$ fibers were found in the dorsal area of the spinal trigeminal nucleus caudalis (Sp5C) and in the ventral area of the spinal trigeminal nucleus interpolaris (Sp5I) (Fig. 4d$_1$–d$_3$); *c-fos* expression was observed in these areas after tFO stimulation (Fig. 4e$_1$–e$_3$, f$_1$–f$_3$). Furthermore, immunohistochemical analysis of phosphorylated ERK (pERK)[43], another neural activity marker, showed EYFP$^+$ fibers surrounding the pERK$^+$ cells in tFO-stimulated reporter mice (Fig. 4g–l). Next, we analyzed the area postrema (AP) and the NTS. The AP and the caudal NTS receive axonal inputs from the VNS, while the rostral NTS from the taste/orofacial tactile nerves[40,41]. We observed EYFP$^+$ fibers in the caudal but not in the rostral NTS in reporter mice (Supplementary Fig. 5); EYFP$^+$ signal was especially strong in the dorsal area

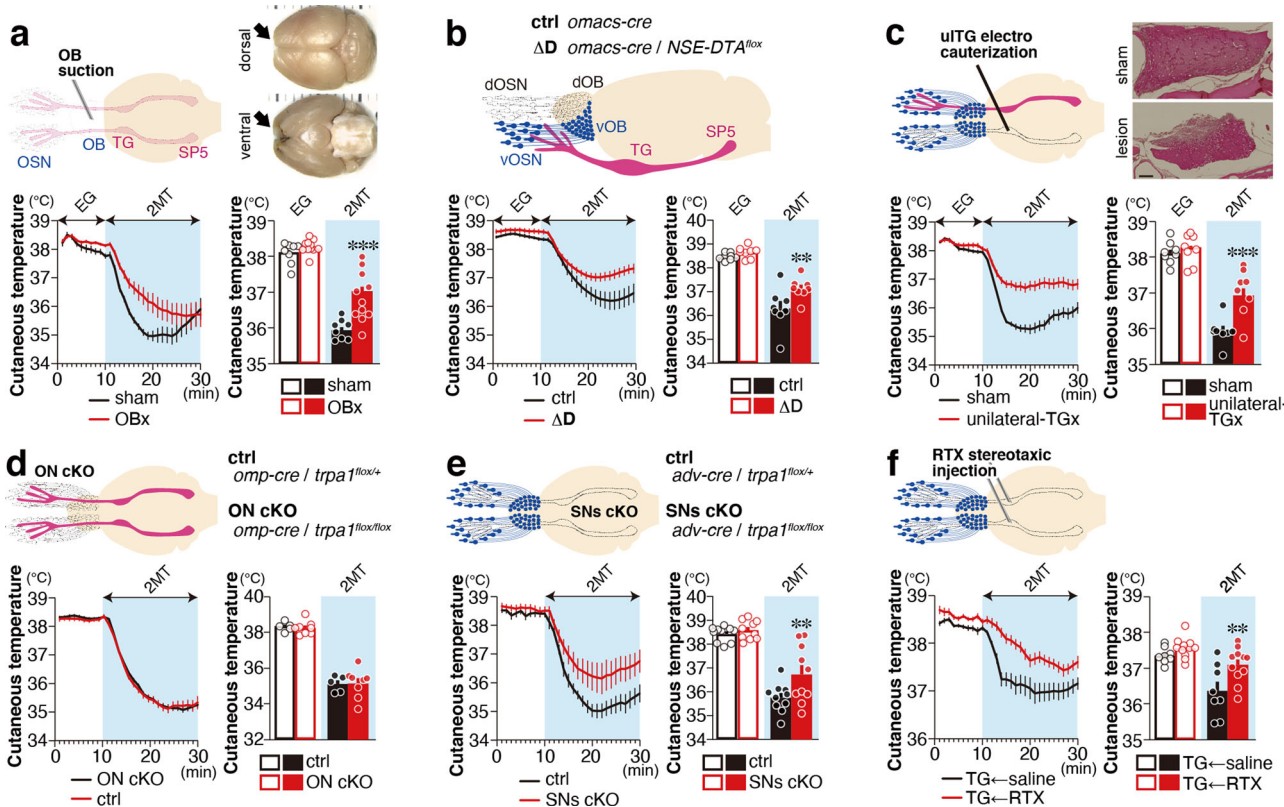

**Fig. 3 Multiple sensory pathways are involved in tFO-induced hypothermia. a** Temporal and mean cutaneous temperature of olfactory bulbectomized (OBx; $n = 12$) and sham-operated ($n = 8$) animals in response to eugenol (EG) and 2MT presentation are shown; $p = 0.6302$ for eugenol presentation (1–10 min) and $p < 0.0001$ for 2MT presentation (11–20 min). **b** Temporal and mean cutaneous temperature of ΔD (*Omacs-Cre; NSE-DTA*$^{flox}$) and control mice ($n = 8$ each) in response to EG and 2MT presentation are shown; $p = 0.722$ for EG presentation (1–10 min) and $p = 0.0052$ for 2MT presentation (21–30 min). **c** Temporal and mean cutaneous temperature of unilateral trigeminal ganglion (TG)-lesioned and sham-operated animals ($n = 8$ each) in response to EG and 2MT presentation are shown; $p = 0.8921$ for EG presentation (1–10 min) and $p < 0.0001$ for 2MT presentation (11–20 min). **d** Temporal and mean cutaneous temperature of olfactory neuron (ON)-specific *Trpa1* knockout (*Omp-Cre; Trpa1*$^{flox/flox}$) and control mice ($n = 5$ for control and $n = 8$ for knockout) in response to 2MT presentation are shown; $p = 0.9757$ for baseline condition (1–10 min) and $p = 0.9943$ for 2MT presentation (21–30 min). **e** Temporal and mean cutaneous temperature of sensory neuron (SN)-specific *Trpa1* knockout (*Adv-Cre; Trpa1*$^{flox/flox}$) and control mice ($n = 10$ each) are shown; $p = 0.9775$ for baseline condition (1–10 min) and $p = 0.0249$ for 2MT presentation (21–30 min). **f** Temporal and mean cutaneous temperature of intra-trigeminal ganglion injection of saline and RTX ($n = 8$ for saline and $n = 11$ for RTX) in response to 2MT presentation are shown; $p = 0.5719$ for baseline condition (1–10 min) and $p = 0.0034$ for 2MT presentation (11–20 min). Data are shown as mean ± SEM. Two-way ANOVA followed by Sidak's multiple comparison test; ** $p < 0.01$; *** $p < 0.001$. Scale bars, 100 μm.

of the caudal NTS. Relatively weak EYFP$^+$ signals were observed in the AP (Fig. 4d$_4$). Meanwhile, *c-fos* expression was observed in the caudal NTS and AP region, where EYFP$^+$ fibers were detected, in tFO-stimulated mice (Fig. 4e$_4$, f$_4$). These results indicate that the information perceived by the *Trpa1*$^+$ neurons in the trigeminal and vagus nerves is transmitted to these specific areas of the brainstem.

**Hypotherima induced by the trigeminal and vagal *Trpa1* pathways.** *Trpa1*-expressing peripheral neurons project their axons to specific areas in the Sp5 and NTS in the brainstem, wherein *c-fos* expression was induced by tFO administration (Fig. 4). We thus asked whether artificial activation of these *Trpa1*$^+$ neurons might induce hypothermia. We first examined the effect of *Trpa1*$^+$ neurons projecting to the Sp5. A Cre-dependent retrograde adeno-associated virus encoding hM3Dq (a chemogenetic activator)[44] fused with mCherry (AAVrg-DIO-hM3Dq) was injected bilaterally in the dorsal and ventral Sp5 of *Trpa1-Cre* and control mice (Fig. 5a). Three weeks after injection, mCherry expression was detected in the TG of *Trpa1-Cre* mice (Fig. 5b). Activation of *Trpa1*$^+$ neurons projecting to the Sp5 by

intraperitoneal (IP) injection of clozapine-N-oxide (CNO) led to increased *c-fos* mRNA expression in the dorsal and ventral Sp5 areas (Fig. 5c). Activation of these neurons by CNO administration induced hypothermia in *Trpa1-Cre* mice, whereas such effect was not observed in control animals or by saline administration (Fig. 5d–f). Next, we analyzed the effects of artificial activation of *Trpa1*$^+$ neurons projecting to the NTS. We injected AAVrg-DIO-hM3Dq into the NTS of *Trpa1-Cre* and control mice (Fig. 6a). Three weeks after infection, we confirmed mCherry expression in the VG (Fig. 6b). Artificial activation of *Trpa1*$^+$ neurons projecting to the NTS by CNO administration induced *c-fos* expression in the NTS (Fig. 6c). CNO administration induced hypothermia in *Trpa1-Cre* mice infected with AAVrg-DIO-hM3Dq into the NTS (Fig. 6d) but not in control mice (Fig. 6e). These results indicate that artificial activation of Sp5 and NTS, which receive axonal projection from *Trpa1*$^+$ neurons, induces hypothermia.

**Diverse activation profiles of TRPA1.** In our model, TRPA1 activation in the TG or VG by tFOs transmits danger information to the Sp5 or NTS to induce multiple physiological effects. It

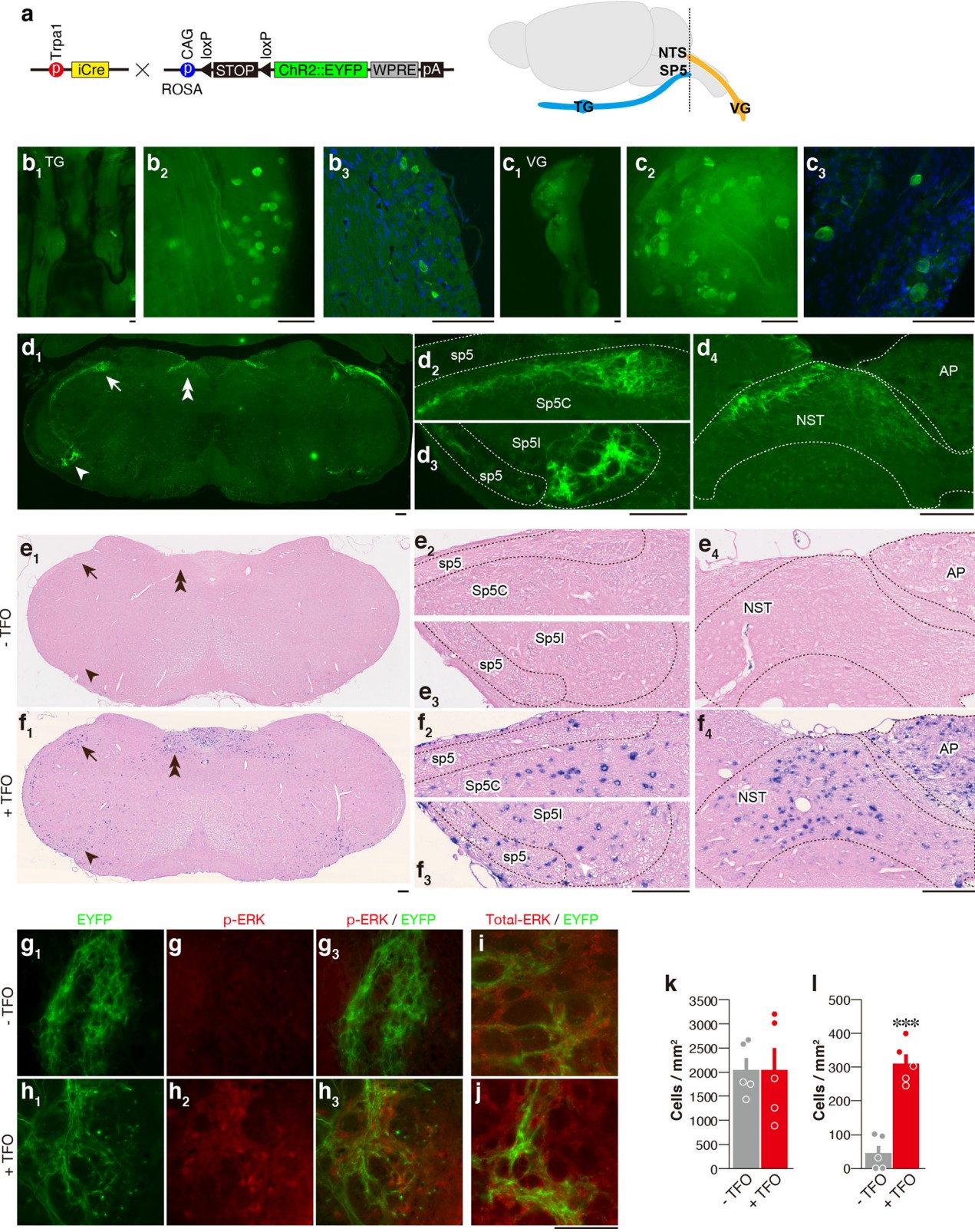

follows that novel compounds inducing more robust effects might be identified by focusing on the molecular activation of TRPA1 and the neuronal activation of TG/VG and Sp5/NTS. We addressed this possibility using known TRPA1 ligands (cinnamaldehyde [CNA] and AITC) and compounds similar to 2MT in chemical structure (4-methyl-2-ethyl-2-thiazoline [4E2MT],

5-methyl-thiazole [5MT], thiomorpholine [TMO], thiophene [TO], and 2-methyl-2-oxazoline [2MO]), for which no anti-hypoxic activity is known (Fig. 7a).

We first performed inside-out patch-clamp recording to analyze ion currents induced by applying these ligands to TRPA1 expressed in HEK293 cells (Fig. 7b). As reported previously[11,45],

**Fig. 4 Projection sites of *Trpa1*+ neurons in the Sp5/NTS. a** Strategy for selective labeling of *Trpa1*+ cells using *Trpa1-Cre* and *RCL-ChR2/EYFP* mice. Schematic illustration of trigeminal and vagus nerve projections to the brainstem are also shown. A dotted line indicates the approximate position of the sections shown in **d–j. b–d** Representative EYFP signals in the TG (**b₁–b₃**), VG (**c₁–c₃**), and medulla (**d₁**) of whole mount views (**b₁, c₁**), magnified whole-mount views (**b₂, c₂**), and tissue sections (**b₃, c₃, d₁**) of *Trpa1-Cre; RCL-ChR2/EYFP* double transgenic mice. Enlarged images of Sp5C (**d₂**; area indicated by arrow in **d₁**), Sp5I (**d₃**; area indicated by arrowhead in **d₁**), and NTS (**d₄**; area indicated by double arrow in **d₁**) are also shown. In the Sp5I/C transition area, YFP-positive fibers were observed in the dorsal (arrow) area in the Sp5C and ventral area in the Sp5I (arrowhead) regions (**d₁**). **e₁–f₄** Representative images of in situ hybridization of *c-fos* RNA in the medulla following IP injection of saline (**e₁–e₄**) and tFO (4E2MT; **f₁–f₄**), along with enlarged images of the Sp5 (**e₂, e₃, f₂,** and **f₃**) and NTS (**e₄, f₄**). **g–l** Expression of phospho-ERK and total ERK was compared with EYFP signals in the ventral EYFP-fiber-rich area in the Sp5 after IP injection of saline (**g, i**) and tFO (4E2MT; **h, j**). Quantification of total ERK (**k**; $n = 5$ for each, $p = 0.9984$) and pERK (**l**; $n = 5$ for each, $p < 0.0001$) are also shown. Data are shown as mean ± SEM. Unpaired, two-tailed Student's *t* test was used to assess significance. Scale bars, 100 μm; ***$p < 0.001$.

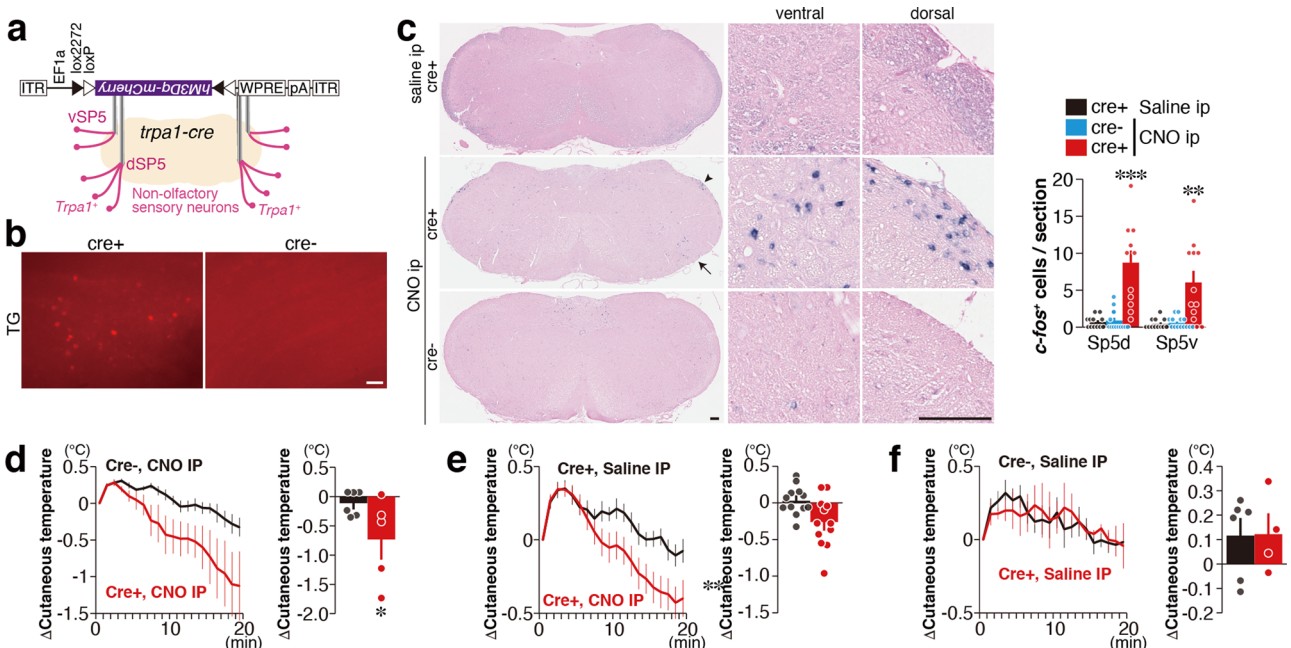

**Fig. 5 Hypothermia induced by the *Trpa1*+ neurons projecting to the Sp5. a, b** The experimental design of the chemogenetic activation of *Trpa1*+ neurons projecting to the Sp5 (**a**) and representative retrograde-labeled mCherry expression in the TG in *Trpa1-Cre* and control mice (**b**). **c** Representative images of medulla (left), magnified views in the ventral and dorsal parts of Sp5 (middle; areas indicated by arrow and arrowhead in the left figure), and quantification (right) of *c-fos*+ cells in the dorsal Sp5 (Sp5d) and ventral Sp5 (Sp5v) after clozapine-N-oxide (CNO) administration (Sp5d, $n = 12$ for *Trpa1-Cre*+/sal, $n = 14$ for *Trpa1-Cre*−/CNO, and $n = 24$ for *Trpa1-Cre*+/CNO, $p = 0.9850$ between *Trpa1-Cre*+/saline and *Trp1-Cre*−/CNO, and $p < 0.0001$ between *Trpa1-Cre*+/saline and *Trpa1-Cre*+/CNO; Sp5v, $n = 12$ for *Trpa1-Cre*+/sal, $n = 13$ for *Trpa1-Cre*−/CNO, and $n = 24$ for *Trpa1-Cre*+/CNO, $p = 0.9947$ between *Trpa1-Cre*+/sal and *Trpa1-Cre*−/CNO, and $p = 0.0043$ between *Trpa1-Cre*+/saline and *Trpa1-Cre*+/CNO) are shown. **d** Temporal and mean cutaneous temperature after CNO administration are shown for hM3Dq-infected *Trpa1-Cre* (red) and control (black) mice ($n = 6$ for control and $n = 5$ for *Trpa1-Cre*, $p = 0.0179$). **e** Temporal and mean cutaneous temperature after administration of saline (black) and CNO (red) are shown for hM3Dq-infected *Trpa1-Cre* mice ($n = 12$ each, $p = 0.0018$). **f** Temporal and mean cutaneous temperature after administration of saline are shown for *Trpa1-Cre* (red) and control (black) mice ($n = 6$ for control and $n = 4$ for *Trpa1-Cre*, $p = 0.500$). Data are shown as mean ± SEM. One-way ANOVA followed by Dunnett's multiple comparison test (**c**), unpaired one-tailed Student's *t* test (**d**), paired one-tailed Student's *t* test (**e**), and Mann–Whitney *U* test (**f**) were used to assess significance; *$p < 0.05$; **$p < 0.01$; ***$p < 0.001$. Scale bars, 100 μm.

HEK293 cells overexpressing TRPA1 demonstrated channel activities in response to CNA and AITC. Likewise, channel activation was also observed in TRPA1-expressing HEK293 cells in response to 2MT. Among the 2MT-like compounds, 4E2MT, TMO, and 5MT, but not 2MO or TO, induced channel activation in these cells.

Next, we performed calcium imaging to analyze the ligand response of *Trpa1*+ TG and VG cells by crossing *Trpa1-Cre* mice with a Cre-dependent GCaMP6-expressing strain[46] (Fig. 7c, d). As in the inside-out patch-clamp analyses, 2MO and TO did not increase calcium influx in *Trpa1*+ TG and VG cells. In contrast, 1 mM CNA and AITC and 10 mM TMO increased calcium influx in both TG and VG *Trpa1*+ cells. While 5MT had the most

robust channel activation response in the inside-out patch-clamp analyses, it did not increase calcium influx in *Trpa1*+ TG and VG cells. 4E2MT led to increased calcium influx in *Trpa1*+ cells only in the VG, whereas 2MT increased calcium influx in those in the TG (Fig. 7c, d). CNA, AITC, and TMO increased calcium influx in a large number of *Trpa1*+ cells, whereas 4E2MT induced a response in only a small number of *Trpa1*+ cells. These results raise the possibility that *Trpa1*+ cells in the VG and TG may have distinct calcium influx profiles to discriminate closely related molecules.

Then, we measured *c-fos* mRNA in the Sp5 and NTS in response to these ligands (Fig. 8a, b). Both vaporized presentation and IP injection of volatile compounds are known to activate

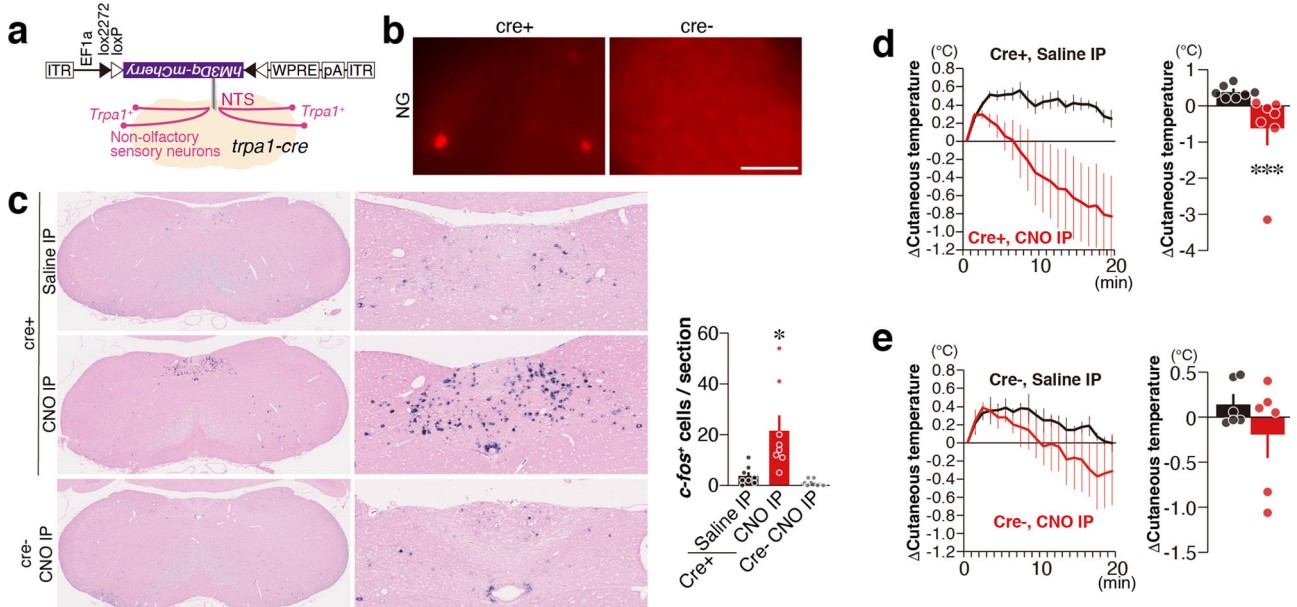

**Fig. 6 Hypothermia induced by the *Trpa1*+ neurons projecting to the NTS. a, b** The experimental design of the chemogenetic activation of *Trpa1*+ neurons projecting to the NTS (**a**) and representative retrograde-labeled mCherry expression in the VG in *Trpa1-Cre* and control mice (**b**). **c** Representative images of medulla (left), magnified views in the NTS (middle), and quantification (right) of *c-fos*+ cells in the NTS after CNO administration are shown ($n = 8$ for *Trpa1-Cre*+/sal and *Trpa1-Cre*+/CNO and $n = 6$ for *Trpa1-Cre*−/CNO, $p = 0.0135$ between *Trpa1-Cre*+/sal and *Trpa1-Cre*+/CNO, and $p = 0.5335$ between *Trpa1-Cre*+/sal and *Trpa1-Cre*−/CNO). **d, e** Temporal and mean cutaneous temperature after administration of saline (black) and CNO (red) are shown for hM3Dq-infected *Trpa1-Cre* (**d**; $n = 7$ each, $p = 0.0003$) and control mice (**e**; $n = 6$ each, $p = 0.1562$). Data are shown as mean ± SEM. Kruskal–Wallis with Dunn's multiple test (**c**), one-tailed, Mann–Whitney $U$ test (**d**), and one-tailed, Wilcoxon test (**e**) were used to assess significance; *$p < 0.05$; **$p < 0.01$; ***$p < 0.001$. Scale bars, 100 μm.

sensory neurons[47,48]. We also previously showed that presentation of tFOs both by vapor and IP injection upregulated *c-fos* expression in the Sp5/NTS and also induced hypothermia and anti-hypoxic effects[5]. Thus, we presented these ligands by IP administration in these experiments. Consistent with the patch-clamp and calcium imaging analyses, 2MO and TO administration did not significantly upregulate *c-fos* in the Sp5 or NTS. Interestingly, among TRPA1 ligands, AITC but not CNA upregulated *c-fos* expression in the Sp5 and NTS. By contrast, all the 2MT-like compounds, other than 2MO and TO, significantly induced *c-fos* expression in the Sp5 and NTS.

*Trpa1*+ sensory neurons project to a particular area in the Sp5/NTS, and 2MT-induced *c-fos* mRNA expression in this area was not observed in *Trpa1*−/− mice[5,9] (Supplementary Fig. 4f), suggesting that activation of TRPA1 by 2MT in the TG/VG might directly induce *c-fos* expression in the Sp5/NTS. However, while 2MT and its structurally related compounds induced *c-fos* expression in the Sp5/NTS, CNA did not, even though the latter activated TRPA1 like the other compounds.

A possible explanation for this discrepancy is that tFOs activate a different population of *Trpa1*+ neurons which are not activated by CNA, and these neurons project to the Sp5/NTS to induce *c-fos* expression. If this model is true, we would find *Trpa1*+ sensory neurons which are activated by tFOs but not by CNA. Accordingly, we compared the population of *Trpa1*+ neurons responding to CNA and those responding to tFOs. The responses for CNA and 4E2MT were simultaneously analyzed in 49 *Trpa1*+ VG cells (Supplementary Fig. 6). However, we found that almost all *Trpa1*+ VG neurons (48/49 neurons) were more strongly activated by CNA than by 4E2MT. The single remaining neuron responded almost equally to CNA and 4E2MT. We did not find any neurons which were activated by 4E2MT alone. Although we

cannot exclude the possibility that a small subpopulation of *Trpa1*+ cells respond to tFOs but not to CNA, our results did not fit the model that *Trpa1*+ sensory neurons dedicated for tFOs induce *c-fos* expression in the Sp5/NTS. Another possibility is that Sp5/NTS *c-fos* expression is induced by non-TG/VG *Trpa1*+ sensory inputs which are relevant to tFOs stimulation but not to CNA stimulation. In the unilateral TG ablated mice, *c-fos* expression in the Sp5 in the lesioned side in response to 2MT presentation was suppressed compared to that in the contralateral side, suggesting that *c-fos* expression in the Sp5 is induced by ipsilateral TG projection neurons (Supplementary Fig. 7). In calcium imaging, 4E2MT activated only *Trpa1*+ VG, but not *Trpa1*+ TG, neurons (Fig. 7c, d). Thus, it is possible that *c-fos* expression in the Sp5/NTS may be induced by 4E2MT, depending on the activation of *Trpa1*+ VG neurons. However, 5MT did not increase calcium influx in most *Trpa1*+ TG/VG neurons (Fig. 7c, d), but it induced *c-fos* expression in the Sp5/NTS (Fig. 8). Thus, we speculate that calcium imaging using isolated TG/VG neurons may not reflect the in vivo response of these cells.

Since tFOs and CNA induce different gene expression in the Sp5/NTS, it is possible that those compounds induce different gene expression also in the TG/VG. To examine this possibility, we compared gene expression in the TG among mice treated with 2MT ($n = 2$), 4E2MT ($n = 4$), TO ($n = 2$), CNA ($n = 4$), AITC ($n = 2$), and saline ($n = 2$) by RNA sequencing (RNAseq). Among the 39,638 genes analyzed, only 12 (0.03%) showed significant differences ($q < 0.01$). Next, we analyzed the correlation among the expressions of these 12 genes and 16 stimulation conditions. Compared to the control (saline) condition, gene expression fluctuations were comparatively smaller in AITC and TO conditions, but larger in 2MT, 4E2MT, and CNA conditions (Fig. 9a). Differentially expressed genes were mainly categorized into two groups. The first group was upregulated in response to

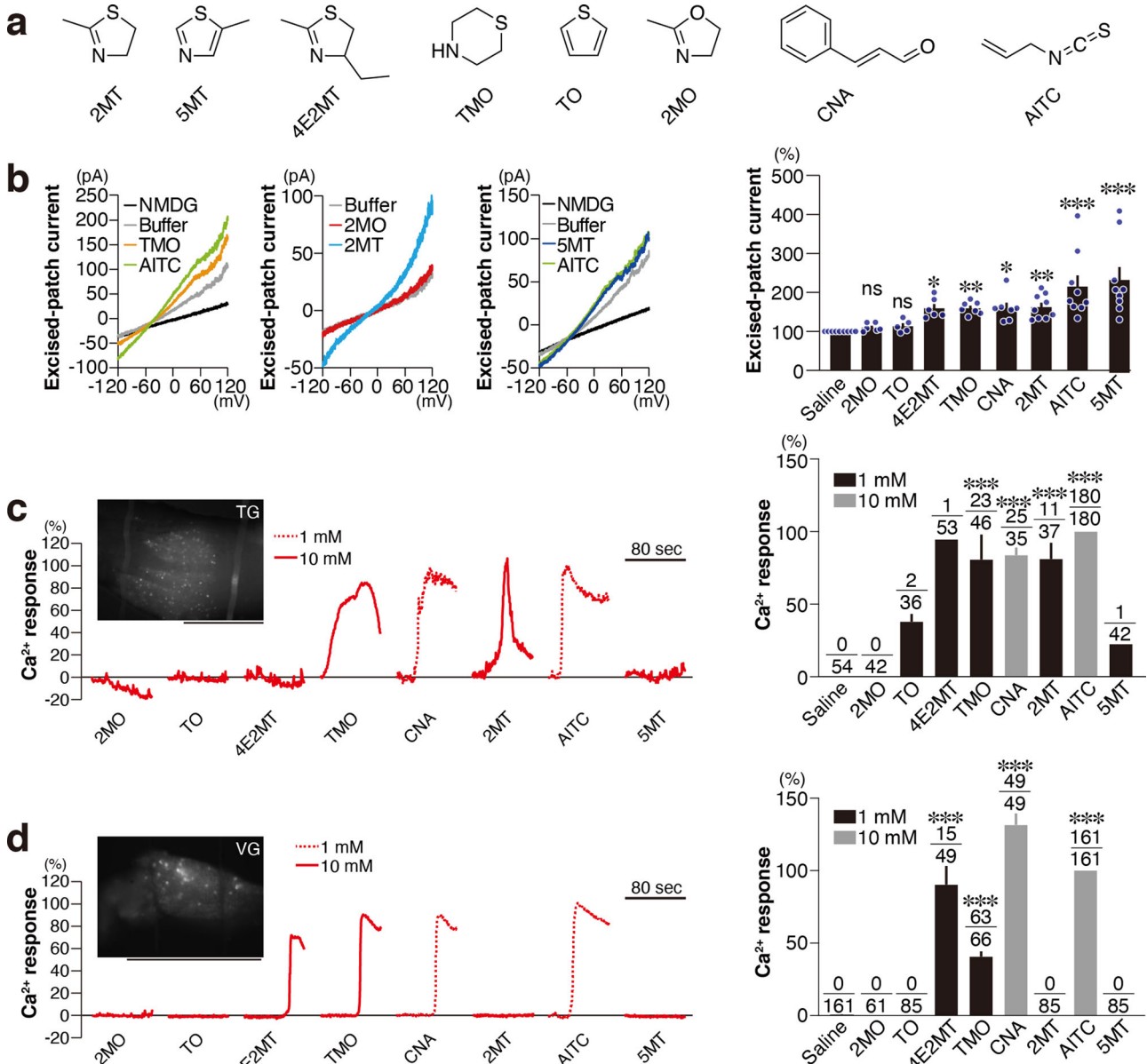

**Fig. 7 TRPA1 activation by various compounds. a** Chemical structures of tested compounds. **b** Representative TRPA1 current–voltage ($I–V$) relations for each compound (left), and relative macroscopic currents from excised patches elicited by application of 100 μM of each compound at +100 mV, normalized to those elicited by the application of saline (right; $n = 5$ for TO, $n = 6$ for 2MO and 4E2MT, $n = 7$ for TMO and CNA, and $n = 9$ for saline, 2MT, AITC, and 5MT; $p > 0.9999$ for 2MO and TO, $p = 0.018$ for 4E2MT, $p = 0.0076$ for TMO, $p = 0.0161$ for CNA, $p = 0.0041$ for 2MT, and $p < 0.0001$ for AITC and 5MT) are shown. **c, d** (Left) Representative traces of GCaMP6f fluorescence of the TG (**c**) and VG (**d**) are shown. A representative image of the TG and VG used in the calcium imaging is also shown. Scale bar, 1 mm. (Right) Calcium activity of AITC-responsive $Trpa1^+$ cells in the TG (**c**) and VG (**d**) in response to the indicated compounds were analyzed and relative calcium activities of responsive cells for each compound are shown. The number of responded and recorded cells are indicated in the bar graphs. Statistical significance was assessed between activities of all the recorded cells in saline condition and responsive cells for each condition (**c**, $p = 0.4916$ for TO, $p = 0.3701$ for 4E2MT, $p < 0.0001$ for TMO, CNA, 2MT, and AITC, and $p = 0.7902$ for 5MT; **d**, $p < 0.0001$ for 4E2MT, TMO, CNA, and AITC). Scale bar, 100 μm. Data are shown as mean ± SEM. Kruskal–Wallis with Dunn's multiple comparison test (**b**) and Kruskal–Wallis with uncorrected Dunn's test (**c, d**) were used to assess significance; *$p < 0.05$; **$p < 0.01$; ***$p < 0.001$.

2MT and its structurally related compound, 4E2MT (Fig. 9b–e). All five genes categorized into this group were immediate early genes (IEGs). The second group was upregulated in response to CNA and 4E2MT (Fig. 9f–i). Six genes were categorized into this group, and half of them are involved in the regulation of the cytoskeleton and extracellular matrix. These results indicate that different compounds induce differential gene expression in the TG. Both 2MT and CNA activated $Trpa1^+$ cells in the TG. Nevertheless, they induced the expression of distinct sets of genes

in the TG. In the current situation, the causal relationship between differential gene expression in the TG and $c$-$fos$ expression in the Sp5/NTS or physiological responses induced by CNA and tFOs are not clarified. It is possible that gene expression in the TG or Sp5/NTS may be useful for prediction of physiological responses of tFOs.

**Identification of novel compounds with ultrapotent anti-hypoxic activities.** 2MO and TO did not activate TRPA1 or

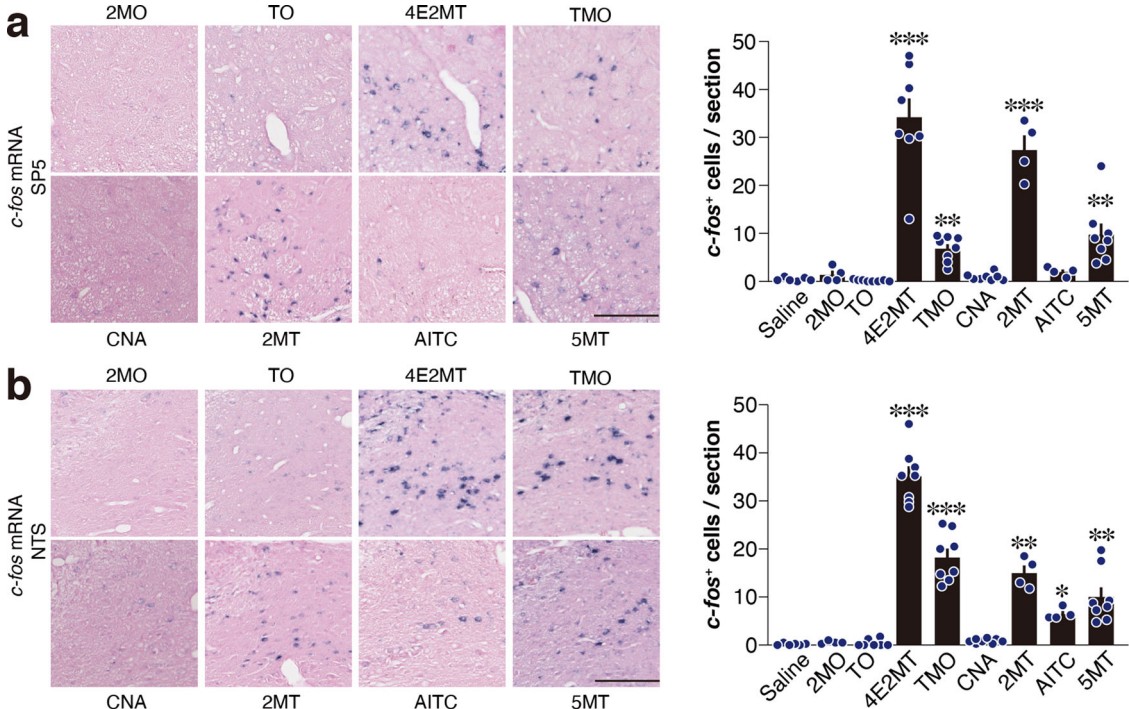

**Fig. 8 Sp5/NTS activation by various compounds. a, b** Representative images of in situ hybridization of *c-fos* mRNA (left) and quantification of *c-fos*-positive cells (right) in the Sp5 (**a**) and NTS (**b**) in response to IP injection of the indicated compounds ($n = 6$ for saline, $n = 4$ for 2MO, 2MT, and AITC, and $n = 8$ for TO, 4E2MT, TMO, CNA, and 5MT; **a**, $p = 0.5447$ for 2MO, $p = 0.5745$ for TO, $p < 0.0001$ for 4E2MT, $p = 0.0071$ for TMO, $p = 0.4904$ for CNA, $p = 0.0005$ for 2MT, $p = 0.2331$ for AITC, and $p = 0.0028$ for 5MT; **b**, $p = 0.5148$ for 2MO, $p = 0.9379$ for TO, $p < 0.0001$ for 4E2MT, $p = 0.0001$ for TMO, $p = 0.2798$ for CNA, $p = 0.0034$ for 2MT, $p = 0.0447$ for AITC, and $p = 0.0043$ for 5MT). Scale bar, 100 μm. Data are shown as mean ± SEM. Kruskal–Wallis with uncorrected Dunn's test was used to assess significance; *$p < 0.05$; **$p < 0.01$; ***$p < 0.001$.

the projection targets of *Trpa1*[+] neurons, Sp5/NTS, in any of the four assays we performed: patch-clamp analysis of TRPA1, calcium imaging of TG/VG, RNAseq analysis of TG, and *c-fos* expression analysis in the Sp5/NTS. By contrast, there were clear differences in the response to known TRPA1 ligands, 2MT, and their structurally related compounds. To determine the assay most predictive of TRPA1-relevant physiological effects, we analyzed the anti-hypoxic effects of these chemicals by IP-injecting them and monitoring survival time in 4% oxygen for up to 30 min. 2MO and TO did not prolong survival time. Importantly, CNA, which activated TRPA1 in the heterologous system and TG/VG cells but did not upregulate IEG genes in TG or *c-fos* in the Sp5/NTS, did not increase survival time, while all other compounds significantly prolonged survival time (Fig. 10a and Supplementary Fig. 8). On the other hand, the present study indicates the involvement of the olfactory system in the regulation of tFO-induced hypothermia. Furthermore, fear-related behaviors to compounds derived from natural predators, e.g., snow leopard urine, in addition to 2MT, TMT, and SBT, are regulated by the olfactory system[6,9]. Thus, it is possible that *c-fos* mRNA expression in the OB could also predict anti-hypoxic activity. To test this possibility, we analyzed the *c-fos* mRNA expression in the OB (Supplementary Fig. 9a–d). 2MT, 4E2MT, and 2MO greatly increased *c-fos* mRNA expression in the OB. Compared with these compounds, the increase in *c-fos* mRNA expression by TO, CNA, and TMO was weaker. Thus, compared to the increase in *c-fos* mRNA expression in the TG/VG, the increase in *c-fos* mRNA expression in the OB was weakly correlated with resistance to hypoxia (Supplementary Fig. 9e). These results indicate that TRPA1 activation in heterologous systems or TG/VG neurons is insufficient; Sp5/NTS activation is also required

to induce anti-hypoxic effects. We further analyzed 4E2MT and TMO as novel ligands showing especially high anti-hypoxic activities.

By prior stimulation with 4E2MT, ATP concentration was maintained in the brain under lethal hypoxic conditions (Supplementary Fig. 10). Whereas 4E2MT stimulation led to robust (>5 °C) hypothermia in *Trpa1*[+/−] mice, this effect was absent in *Trpa1*[−/−] mice (Fig. 10b), as was the prolongation of survival time in 4% oxygen (Fig. 10c).

TMO was also highly effective in inducing survival in hypoxic conditions: all mice administered TMO survived more than 30 min in 4% oxygen. To further analyze the extent to which TMO could prolong survival in hypoxic conditions, we measured the length of time each mouse remained alive in hypoxic conditions (Fig. 10d). 2MT stimulation significantly increased survival time in hypoxic conditions compared to control conditions[5]. Compared to 2MT stimulation, TMO stimulation dramatically prolonged survival time in the 4% oxygen condition. Accordingly, TMO stimulation suppressed oxygen consumption more prominently than 2MT stimulation (Fig. 8e). Taken together, these results indicate that potent inducers of TRPA1-relevant physiological effects, including hypothermia, hypometabolism, and hypoxia resistance, can be identified by analyzing the activities of TRPA1 and Sp5/NTS (Supplementary Fig. 11).

## Discussion
We focused on *Trpa1* as a receptor gene responsible for tFO-induced physiological effects. By a large-scale forward genetic screening, we identified *Trpa1* as a novel target gene inducing freezing behavior in response to 2MT, a type of tFO. *Trpa1* regulated not only fear-related behaviors induced by 2MT, but also those induced by a component of fox secretion, TMT, and

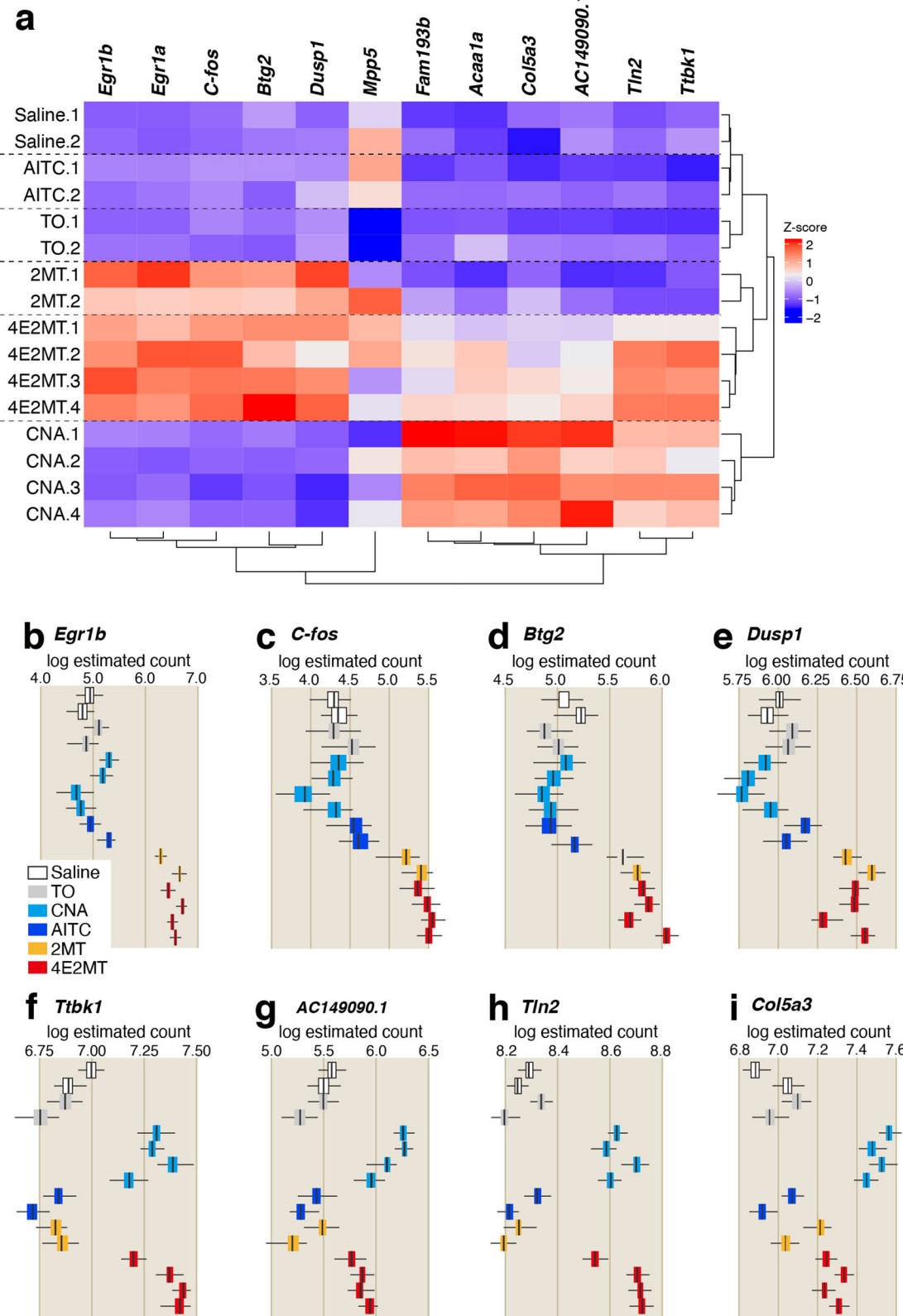

**Fig. 9 Compounds induce differential gene expression in the TG. a** Clustered heat map depicting relationships across 12 genes, which exhibited significant differences ($q < 0.01$) under 16 stimulation conditions. The color bar indicates the z score scale. *Egr1a* and *Egr1b* are Ensemble transcript ENSMUST0000006479.5 and ENSMUST00000165033.1, respectively. **b–i** Box plots depicting the log estimated counts of gene expression in 16 stimulation conditions are shown for the indicated genes that exhibited significant differences ($q < 0.01$); $n = 2$ for saline, TO, AITC, and 2MT, and $n = 4$ for CNA and 4E2M; $q = 0.00024$ for *Egr1b* (**b**), $q = 0.0012$ for *c-fos* (**c**), $q = 0.0024$ for *Btg2* (**d**), $q = 0.0067$ for *Dusp1* (**e**), $q = 0.0047$ for *Tbk1* (**f**), $q = 0.0045$ for *AC149090.1* (**g**), $q = 0.0058$ for *Tln2* (**h**), and $q = 0.0083$ for *Col5a3* (**i**). All boxplots indicate median (center line), 25th and 75th percentiles (bounds of box), and minimum and maximum (whiskers); $q$ value obtained by likelihood ratio test was corrected by Benjamin–Hochberg multiple test.

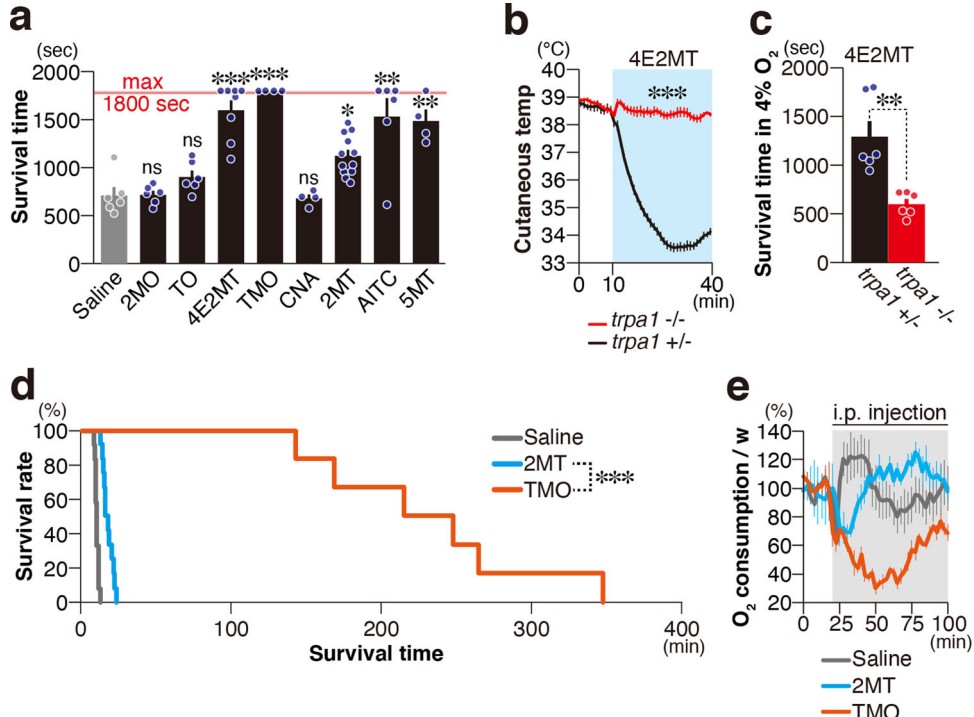

**Fig. 10 Identification of novel compounds with ultrapotent anti-hypoxic activities. a** Mean survival time in 4% oxygen in response to IP administration of the indicated compounds are shown ($n = 6$ for saline, 2MT, TO, and AITC; $n = 8$ for 4E2MT; $n = 4$ for TMO, CNA, and 5MT; and $n = 12$ for 2MT; $p = 0.8728$ for 2MO, $p = 0.2701$ for TO, $p = 0.0001$ for 4E2MT and TMO, $p = 0.9493$ for CNA; $p = 0.0182$ for 2MT; $p = 0.0015$ for AITC; and $p = 0.0036$ for 5MT, on Kruskal–Wallis with uncorrected Dunn's test). **b** Temporal analysis of cutaneous temperature in $Trpa1^{-/-}$ (red) and control (black) mice in response to IP administration of 4E2MT are shown ($n = 6$ for each genotype). Statistical significance was assessed for cutaneous temperature after IP administration of 4E2MT (11–40 min; $p < 0.0001$, Student's $t$ test, unpaired, one-tailed). **c** Mean survival time in 4% oxygen in $Trpa1^{-/-}$ (red) and control (black) mice in response to IP administration of 4E2MT ($n = 6$ for each genotype, $p = 0.0011$, one-tailed Mann–Whitney $U$ test). **d** Survival rate in 4% oxygen with prior IP administration of saline (gray; $n = 6$), 2MT (blue; $n = 12$), and TMO (orange; $n = 6$). **e** Temporal analysis of oxygen consumption in response to IP administration of saline (gray), 2MT (blue), and TMO (orange) ($n = 4$ for each condition). Data are shown as mean ± SEM; *$p < 0.05$; **$p < 0.01$; ***$p < 0.001$.

snake skins[9]. Moreover, we demonstrated that *Trpa1* is involved in tFO-induced physiological responses including hypothermia, hypometabolism, and hypoxia resistance. We propose a model in which *Trpa1* is a central gene in fear perception, responsible for the integrative regulation of behavioral and physiological survival-enhancing responses. On the other hand, tFOs induce fear-related behaviors via the olfactory pathway[6,8]. The present study also indicates that the olfactory pathway is involved in tFO-induced physiological responses. It is possible that in regulating tFO-induced behavioral and physiological responses, there is some separation between the roles of the olfactory pathway mediated by odorant receptors and the trigeminal/vagal pathway mediated by *Trpa1*. However, their differential roles are currently unknown.

2MT stimulation mitigates hypoxic damages and ischemia-reperfusion injuries[5]. In the present study, we showed that hypothermia, hypometabolism, and hypoxia resistance induced by 2MT and other tFOs are partially regulated by *Trpa1* in the trigeminal and vagal nerves. Electrical stimulation of the VNS yields anti-inflammatory effects, and its clinical applications have been proposed. It is considered that VNS yields anti-inflammatory effects via artificially intervening inflammatory reflex system, which maintains homeostasis of immune response[49]. Moreover, VNS has therapeutic effects on drug-resistant epilepsy and depression. Through its anti-inflammatory responses, it is also expected to have therapeutic effects on a wide spectrum of diseases, including sepsis, cardiovascular disease, traumatic brain injury, stroke, and diabetes[50]. TFOs activate VG

neurons and NTS; thus, tFO-stimulation possibly has therapeutic effects comparable to those of VNS. Whether the combination of tFO–TRPA1 has therapeutic effects on various diseases should be examined in a future study.

Different TRPA1-activating ligands induce different physiological responses. For example, AITC, a pungent component of wasabi and mustard oil, induces pain, whereas miR-711 secreted by skin lymphoma induces itch[15,51]. Conversely, acetaminophen and $\Delta^9$-tetrahydrocannabinol have antinociceptive effects via *Trpa1* activation[25,26]. *Trpa1* activation is known to be involved in pro-inflammatory responses[13,16,45,52]. On the other hand, *Trpa1* agonist CNA was recently reported to have anti-inflammatory effects[53]. These contrasting findings indicate that *Trpa1* induces either positive or negative effects on nociception and inflammatory responses, depending on the ligand stimulus. We demonstrated that CNA induced ion currents in *Trpa1*-transfected HEK293 cells and calcium influx in $Trpa1^+$ TG and VG cells, consistent with previous studies[45]; nevertheless CNA demonstrated low anti-hypoxic effects. For clinical applications, it is important to identify novel ligands capable of inducing TRPA1-inducing anti-hypoxic effects. Our data indicate that sole induction of ion currents via TRPA1 is not sufficient to induce anti-hypoxic effects; the induction of neural activation markers in the Sp5/NTS is also necessary. If this principle can be applied to other organisms, including humans, this strategy can be used to screen novel drugs promoting anti-hypoxic effects in clinical applications.

## Methods

**Mice.** Male C57BL/6NCr mice were purchased from Japan SLC, Inc (Shizuoka, Japan). The *Trpa1*$^{-/-}$ mice (stock number 006401), *Trpv1*$^{-/-}$ mice (stock number 003770), *Trpa1*$^{flox}$ mice (stock number 008649), *Omp-Cre* mice (stock number 006668), *RCL-GCamP6f* mice (stock number 028865), and *RCL-ChR2(H134R)/EYFP* mice (stock number 024109) were purchased from The Jackson Laboratory (Bar Harbor, ME, USA). *Trpa1-Cre*$^+$ mice were generated by inserting improved-Cre[54] at the start codon of the endogenous *Trpa1* locus using CRISPR/Cas9-mediated genome editing in the ES cells[55]. *RCL-ChR2(H134R)/EYFP* mice were crossed with *Trpa1-Cre*$^+$ mice to visualize *Trpa1-Cre*$^+$ cells. The ΔD mice were generated by crossing *OMACS-Cre* and *Eno2-STOP-DTA* mice, as reported previously[6]. Advillin-Cre mice were kindly provided by Dr. Wang[31]. Mice were housed under a standard 12-h light/dark cycle at room temperature of $23 \pm 2\,°C$ with humidity of 30–70% and allowed ad libitum access to food and water. Mice were at least 9 weeks old at the start of testing. The experimental protocols were approved by the Animal Research Committee of Kansai Medical University.

**Chemical compounds.** We purchased 2MT, 4E2MT, 5MT, TMO, TO, 2MO, CNA, and AITC from Tokyo Chemical Industry Co., Ltd. (Tokyo, Japan). The vaporized compound presentation was performed by introducing a piece of filter paper containing 271 μmol of a test compound into the test cage. Administration of the compound was performed by intraperitoneally injecting 100 μl of 1% solution in saline (~40 mg/kg). 2MT and TMO can be dissolved in saline at 1%. For the other compounds, because of low solubility in saline, 1% of the compound was added to saline and stirred vigorously by vortex to ensure that the compound was dispersed in the saline just prior to IP administration. Presentation and administration of chemical compounds were performed in an open cage placed in a chemical fume hood to avoid cross-contamination.

**Measurement of cutaneous temperature.** For measurement of cutaneous temperature, mice were anesthetized with pentobarbital (50 mg/kg, IP) 2–3 days prior to testing, and the fur on the back was removed with a chemical hair remover. Each mouse was placed in a separate test cage (17.5 × 10.5 × 15 cm) without a lid placed in a chemical fume hood, habituated for 10 min, and subjected to presentations of filter paper containing test compounds (Figs. 1, 2a, b, 3a–f, and Supplementary Figs. 1–4) or IP administration (Figs. 4, 6e, f, 7, 8, and Supplementary Figs. 6–8) of test compounds. Cutaneous temperature was recorded using an infrared digital thermographic camera (H2640; NEC Avio Infrared Technologies Co., Ltd., Tokyo, Japan) and TH92–707 data capture program 1.2(J) (NEC Avio Infrared Technologies Co., Ltd.) at 5 frames/s. Cutaneous temperature on the back was automatically analyzed using specially designed software based on a previously reported method[56].

**Measurement of core body temeperature and heart rates.** Measurement of core body temperature and heart rate was performed using a radio-telemetry transmitter (TA11ETA-F10; DataSciences International, St Paul, MN, USA) and Dataquest A.R.T. 4.30 (DataSciences International) according to a method[5]. Briefly, a radio-telemetry transmitter was implanted following the surgical procedure described by the manufacturer. After the surgery, the mice were allowed to recover for ~10 days before testing. On the test day, each mouse was placed in a separate test cage (17.5 × 10.5 × 15 cm) without a lid placed in a chemical fume hood, habituated for 10 min, and subjected to presentation of filter paper containing test compound (Fig. 1b, e and Supplementary Fig. 1).

For the restrained condition, mice were implanted with radio-telemetry probes approximately 10 days prior to the experiment, as described above. On the test day, each mouse was placed in a separate test cage (17.5 × 10.5 × 15 cm) without a lid, allowed to habituate for 10 min, and physiological parameters were analyzed for 10 min. Immediately afterward, mice were restrained in ventilated 50-ml plastic tubes (Becton Dickinson and Company, Franklin Lakes, NJ, USA), and physiological parameters were analyzed for 30 min (Fig. 1d, f).

Physiological parameters were automatically transmitted from the device every 10 s using Dataquest A.R.T. software (DataSciences International).

**Measurement of oxygen consumption.** Measurement of oxygen consumption was analyzed using a mass spectrometric calorimeter (ARCO-2000, ARCO System, Chiba, Japan) according to a method[5]. Mice were introduced into a metabolic chamber and habituated for more than 60 min. Following habituation, oxygen consumption was measured for 10 min. Afterward, two pieces of filter paper dropped with 25.7 μl of 2MT were presented (Fig. 2a) or 100 μl of 1% test compound (~40 mg/kg) in saline was IP injected (Fig. 8e), and oxygen consumption was measured.

**Hypoxia resistance.** Hypoxia resistance was analyzed according to a method[5]. For Fig. 2b, mice were presented with a filter paper containing 25.7 μl of 2MT in an open cage with a stainless-steel wire bar lid (26 × 18 × 14 cm). Ten minutes after presentation, the mice were moved to a separate test chamber (17 × 17 × 18.5 cm) supplemented with 4% oxygen. Each chamber had two holes on opposite sides at different heights (4.5 cm and 12 cm, respectively) and a wire mesh platform (height: 9 cm) where mice were confined for the duration of the experiment. To

produce an environment with 4% $O_2$, compressed nitrogen gas and compressed air cylinders were connected to two gas permeators (PD-1B-2; Gastec Corp., Kanagawa, Japan). A mixture of 1600 ml/min nitrogen gas and 400 ml/min air was poured into the test chamber through the upper hole.

For Figs. 2c, d and 8d, mice were intraperitoneally injected with 100 μl of 1% test compound and introduced into separate cages. Thirty minutes after injection, the mice were moved to a separate test chamber (17 × 17 × 18.5 cm) supplemented with 4% oxygen.

For Supplementary Fig. 6, mice were intraperitoneally injected with 200 μl of concentrations (0.01%, 0.1%, 1%, and 10%) of CNA in saline. Thirty minutes after injection, the mice were moved to a separate test chamber (17 × 17 × 18.5 cm) supplemented with 4% oxygen.

The Δ$^9$-THC was kindly provided by Ruri Hanajiri and acetaminophen was purchased from TCI.

**IEG mapping.** IEG mapping was conducted by performing in situ hybridization using antisense RNA probes for *c-fos*, according to a method[5]. For Figs. 4e, f, 6e, f, and Supplementary Fig. 7, C57BL/6 mice were introduced into a separate cage (29 × 19 × 13 cm) with a metal with a stainless-steel wire bar lid and habituated for 2 h. Following habituation, mice were intraperitoneally injected with 100 μl of saline or 1% solution of indicated compound in saline (~40 mg/kg). After 30 min of IP injection, mice were sacrificed, and coronal brain sections were prepared.

For Fig. 5c, i, AAV-injected animals were introduced into a separate cage (29 × 19 × 13 cm) with a stainless-steel wire bar lid and habituated for 2 h. Following habituation, mice were intraperitoneally injected with 5 mg/kg of CNO (Sigma-Aldrich) or saline. After 30 min of IP injection, mice were sacrificed, and coronal brain sections were prepared.

For Supplementary Fig. 4, *Trpa1*$^{-/-}$ and control mice were introduced into a separate cage (29 × 19 × 13 cm) with a metal with a stainless-steel wire bar lid and habituated for 2 h. After habituation, a filter paper dropped with 25.7 μl of saline or 2MT was presented every 5 min for a 30-min period. Following 30 min of presentation, mice were sacrificed, and coronal brain sections were prepared.

For preparing brain sections, mice were anesthetized with gaseous isoflurane (Mylan, Canonsburg, PA) and perfused with ice-cold 4% paraformaldehyde (PFA) in phosphate-buffered saline (PBS). The brains were then removed and immersed in 4% PFA in PBS overnight at 4 °C. The fixed brains were dehydrated in a graded ethanol and xylene series and then embedded in paraffin using an automated system (Sakura rotary, RH-12DM; Sakura Finetek, Tokyo, Japan). Coronal sections with a thickness of 5 μm were prepared using an automatic slide preparation system (AS-200S, Kurabo, Osaka, Japan).

In situ hybridization was performed using an automated system (Discovery XT, Ventana Medical Systems, Oro Valley, AZ) according to the manufacture's protocol. The digoxigenin (DIG; Roche, Germany) labeled antisense RNA probes for *c-fos* were prepared from the plasmid containing DNA fragments spanning the 129 to 537 and the 543 to 1152 bp regions of mouse *c-fos* according to the manufacture's protocol. The DIG-labeled probes (1:1000 dilution) were hybridized for 3 h using a RiboMap Kit (Roche) at 74 °C. The slides were then incubated with biotin conjugated anti-DIG antibody (1:500, Jackson ImmunoResearch, West Grove, PA) at 37 °C for 28 min. The probe was detected using the Ventana BlueMap Kit (Roche, Basel, Switzerland) at 37 °C for 6 h, and counterstained with a Red counterstain kit (Roche) at 37 °C for 4 min. Coverslips were applied using an automated system (Tissue Tek® GlasTM; Sakura Finetek). The stained images were scanned using a NanoZoomer virtual microscope system (2.0 RS, NDP.scan 2.5, and NDP. View2, Hamamatsu Photonics, Hamamatsu, Japan). The number of *c-fos*$^+$ cells in the stained images was then counted by single-blinded investigators.

**Histology.** For whole-mount analysis of *Trpa1-Cre*$^+$ cells in TG and VG, EYFP signals from *Trpa1-Cre/RCL-ChR2(H134R)/EYFP* were observed using a BZ-9000 fluorescence microscope using BZ-II software (Keyence, Osaka, Japan).

For the immunohistochemical analysis of *Trpa1-Cre*$^+$ cells, coronal paraffinized sections were prepared from *Trpa1-Cre/RCL-ChR2(H134R)/EYFP* mice and deparaffinized with xylene followed by rehydration with a graded ethanol series. After incubation in blocking buffer (5% goat serum/0.3% Triton X-100/PBS) for 30 min at room temperature, the slides were incubated with anti-GFP (1:1000, Abcam, Cambridge, MS) in blocking buffer for overnight at 4 °C, followed by incubation with anti-rabbit antibody conjugated with Alexa fluor 488 (1:800, Invitrogen, Carlsbad, CA) for 1.5 h.

For the analysis of phospho-ERK (p-ERK), *Trpa1-Cre/RCL-ChR2(H134R)/EYFP* or *Trpa1-Cre/RCL-GCamP6f* mice were habituated for 2 h in a test cage, then 1% 4E2MT or saline was injected (100 μl, IP; ~40 mg/kg). Four minutes after the injections, mice were perfused with 4% PFA in PBS, and brains were harvested. Brains were post-fixed and immersed in 30% sucrose/PBS overnight, then embedded in OCT compound; 30-μm-thick sections were incubated in blocking buffer (5% goat serum/0.3% Triton X-100/PBS) for 30 min at room temperature, then incubated with anti-p-ERK (1:200, Cell Signaling Technology, Tokyo, Japan) and anti-GFP (1:1000, nacalai tesque, Kyoto, Japan) in blocking buffer for overnight at 4 °C, followed by incubation with anti-rabbit antibody conjugated with Cy3 (1:800, Jackson ImmunoResearch, West Grove, PA) and anti-rat antibody conjugated with Alexa Fluor 488 (1:800, Jackson ImmunoResearch, West Grove,

PA, USA) for 1.5 h. Slides were covered with DAPI-containing mounting medium (Vector Laboratories, Burlingame, CA, USA), and fluorescent images were obtained with a DMI6000 B microscope using LAS AF software (Leica, Wetzlar, Germany).

**Surgery**. Olfactory bulbectomy was performed as described previously. Briefly, mice were anesthetized via an IP injection of pentobarbital (50 mg/kg) and bilateral OBs were removed by aspiration through a glass pipette[57]. The unilateral lesion of the TG was performed as described previously[9].

**RTX injection**. For the injection of RTX into the TG, RTX (100 ng/μl, Sigma Aldrich Corp, St. Louis, MO, USA) was infused bilaterally into the TG of C57BL/6NCr mice (coordinates: AP, −0.1 mm; LR, ±1.1 mm, ±1.5 mm; DV, −6.5 mm from the bregma, in total four sites/mouse, 0.5 μl/site) using a 10-μl Hamilton syringe mounted on an UltraMicroPump (UMP3; World Precision Instruments LLC, Sarasota, FL, USA) and its controller (Micro4, World Precision Instruments LLC). As a control, saline was infused into the same coordinates ($n = 6$ each). After 1–2 weeks of recovery, the hair on the back was removed and cutaneous temperature was measured as described above.

**hM3Dq activation of *Trpa1*$^+$ cells projecting to the Sp5 and NTS**. Mice were anesthetized and placed on a stereotaxic device (Narishige, Tokyo, Japan) with the head bent downward. An incision was made in the skin of the dorsal neck and muscles were dissected to reveal the membrane overlying the dorsal medulla. A retrograde AAV virus carrying a double-floxed inverted hM3D(Gq) gene (AAVrg-hSyn1-DIO-hM3D(Gq)-mCherry, Addgene) was infused into the Sp5 or NTS using a glass pipette connected to a Nanoject III (Drummond Scientific Co, Broomall, PA, USA). Injections into ths Sp5 were made bilaterally at a rate of 2 nl/s at the following coordinates: −0.5 mm caudal and ±1.6 mm lateral from the caudal end of the cerebellum at a depth of 0.5 mm and 1.1 mm from the surface (in total 4 sites/mouse and 0.5 μl/site). Injections into the NTS were made bilaterally at a rate of 1 nl/s at the following coordinates: −0.3 mm caudal and ±0.2 mm lateral from the caudal end of the cerebellum at a depth of 0.5 mm from the surface (in total 2 sites/mouse and 0.08 μl/site). The hair of the back was removed after at least 3 weeks of recovery, and cutaneous temperature was measured 2–3 days later.

On the test day, mice were intraperitoneally administered with saline or 5 mg/kg of CNO (Sigma-Aldrich) after 10 min of habituation in a test cage (17.5 × 10.5 × 15 cm) without a lid. Cutaneous temperature was measured for 20 min.

For *c-fos* quantification, mice received a saline or CNO injection (5 mg/kg, IP) and transferred to a 4% oxygen chamber 30 min later. Mice were perfused with 4% PFA/PBS after they stopped breathing. The brain was paraffinized and *c-fos* in situ hybridization was performed as described above.

**Electrophysiology**. HEK293 cells (RCB1637) were obtained from the RIKEN BRC Cell bank. Cells were maintained in Dulbecco's modified Eagle's medium (DMEM) supplemented with 10% FBS and 1% penicillin–streptomycin at 37 °C with 5% $CO_2$. To obtain HEK293 cells stably expressing TRPA1, cells were transfected with pTRPA1-P2A-mCherry, which express mouse TRPA1 together with mCherry[9], using Lipofectamine 2000 reagent. Twenty-four hours after transfection, the medium was replaced with DMEM containing 400 μg/ml neomycin. After selection, 30 independent colonies were picked, and mCherry-expressing cells were selected for electrophysiological analysis.

Electrophysiological analysis was performed according to a previous study[58]. Briefly, the day before electrophysiological analysis, ~2 × 10$^5$ cells were seeded into a 35-mm dish. Stably transfected HEK293 cells were subjected to inside-out patch voltage clamp voltage experiments using an EPC800 USB patch-clamp amplifier (HEKA Instruments Inc., Holliston, MA, USA).

The ramping protocol consisted of a 1200-ms ramp from −120 to +120 mV from a holding potential of 0 mV applied every 10 s. Inside-out patches were recorded with a *N*-methyl-D-glucamine (NMDG) solution in the patch pipette: 150 mM NMDG, 150 mM HCl, 5 mM EGTA, 0.61 mM $MgCl_2$ to obtain 0.5 mM free $Mg^{2+}$, 0.13 mM $CaCl_2$ to obtain 10 nM free $Ca^{2+}$ (calculated using maxchelator, http://maxchelator.stanford.edu), and 5 mM HEPES, pH 7.4. Compounds were dissolved in low $Ca^{2+}$ solution, that was also used for perfusion of the intracellular side of the patch, containing: 150 mM NaCl, 5 mM EGTA, 0.61 mM $MgCl_2$ to obtain 0.5 mM free $Mg^{2+}$, 0.13 mM $CaCl_2$ to obtain 10 nM free $Ca^{2+}$, and 5 mM HEPES, pH 7.4. To confirm the quality of the recordings, we regularly substituted all cations of the intracellular side solution with $NMDG^+$, which is expected to reduce the current to nearly background levels[59]. Experiments were performed at a constant temperature (>25 °C).

**Calcium imaging of isolated TG and VG**

*Animals and imaging session.* TrpA1-Cre/RCL-GCamP6f mice (5–12 weeks old) were decapitated, and their TG or VG were isolated and transferred in oxygenated extracellular solution containing (in mM): 20 HEPES, 124 NaCl, 1.8 KCl, 1.24 $KH_2PO_4$, 2 $MgCl_2$, 2 $CaCl_2$, 10 D-glucose, pH 7.4. TG or VG was mounted onto a closed chamber and perfused with the extracellular solution or test solution at a rate of 2 ml/min. TG or VG was first stimulated with 1 mM AITC and then with

other agonists sequentially. Each stimulation lasted 25 s, and the intervals were at least 150 s. Imaging was performed with an epifluorescence microscope (MVX10, Olympus, Tokyo, Japan). GCaMP6f-expressing neurons were excited with 460–480 nm light from a mercury arc lamp (U-LH100HG, Olympus), and the fluorescence was imaged through a 495–540 nm band-pass filter. Images were acquired with a CMOS camera (acA2040–55um; Basler AG, Ahrensburg, Germany) and Pylon5 software (Basler AG) at 2 Hz.

*Data analysis.* Fluorescence data were analyzed using ImageJ 1.52 and Excel 2010. Movies were spatially down-sampled by a factor of 4. Regions of interest (ROIs) of AITC-activated cells were manually drawn around the cell bodies. Artificial trends mainly caused by photobleaching were compensated by fitting the baseline recording period with single exponential curve. Calcium activity was calculated by subtracting the baseline activity (mean fluorescence during 15 s immediately before application of the agonists) from evoked activity (mean fluorescence during 5 s before and after the peak). Response to AITC (1 mM) was used as a positive control for *Trpa1*$^+$ cells. For each test compound, responsive cells were defined as cells showing at least one peak higher than the level of negative control (saline).

**RNA sequencing**. C57BL/6 mice were habituated for 2 h in a test cage, then intraperitoneally injected with 100 μl of 1% of the test compound in saline (~40 mg/kg). Following 30 min of IP injection of the test compound, trigeminal ganglia were dissected, and RNA was extracted using RNeasy Mini Kit (Qiagen) according to the manufacturer's protocol. Library construction and sequencing were performed by Macrogen (Kyoto, Japan). Raw reads were trimmed by Trimmomatic v0.36 (ref. [60]) with 'LEADING:30 TRAILING:30 SLIDINGWINDOW:4:15 MIN-LEN:60' options to keep high-quality sequences, and transcript abundances were quantified by Kallisto v0.44.0 (ref. [61]) with 100 bootstraps using the reference transcripts on mouse genome GRCm38 (Ensembl release 93). Differentially expressed transcripts were identified using likelihood ratio test of Sleuth R package v0.30.0 (ref. [62]) with *q*-value of <0.01.

**Statistics and reproducibility**. GraphPad Prism 8 and Microsoft Excel for Mac (version 16.44) were used for statistical analysis. The statistical methods used for each experiment are listed below. The significance level for all the tests was set at $p < 0.05$.
Figure 1a; D'Agostino-Pearson omnibus test, followed by two-way ANOVA with Sidak's multiple comparison test, was used to assess between *Trpa1*$^{−/−}$ and *Trpa1*$^{+/−}$ for cutaneous temperature before and during 2MT presentation. Figure 1b; D'Agostino-Pearson omnibus test, followed by two-way ANOVA with Sidak's multiple comparison test, was used to assess between *Trpa1*$^{−/−}$ and *Trpa1*$^{+/−}$ for core temperature before and during 2MT presentation. Figure 1c; D'Agostino-Pearson omnibus test, followed by two-way ANOVA with Sidak's multiple comparison test, was used to assess between *Trpv1*$^{−/−}$ and *Trpv1*$^{+/−}$ for cutaneous temperature before and during 2MT presentation. Figure 1d; D'Agostino-Pearson omnibus test, followed by two-way ANOVA with Sidak's multiple comparison test, was used to assess between *Trpa1*$^{−/−}$ and *Trpa1*$^{+/−}$ for cutaneous temperature before and during restrained condition. Figure 1e; Anderson-Darling, Shapiro–Wilk, and Kolmogorov–Smirnov tests, followed by two-way ANOVA with Sidak's multiple comparison test, were used to assess between *Trpa1*$^{−/−}$ and *Trpa1*$^{+/−}$ for heart rates before and during 2MT presentation. Figure 1f; D'Agostino-Pearson omnibus test, followed by two-way ANOVA with Sidak's multiple comparison test, was used to assess between *Trpa1*$^{−/−}$ and *Trpa1*$^{+/−}$ for heart rate before and during 2MT presentation. Figure 2b; the log-rank test was used to assess between *Trpa1*$^{−/−}$ and *Trpa1*$^{+/−}$ for survival time in 4% oxygen. Figure 2c; the log-rank test was used to assess between saline and Δ9-THC administrations, between saline and AITC administrations, and between saline and APAP administrations for survival time in 4% oxygen. Figure 2d; the log-rank test was used to assess between prior IP administration of saline and APAP for *Trpa1*$^{−/−}$ and *Trpa1*$^{+/−}$, respectvely. Figure 3a; D'Agostino-Pearson omnibus test, followed by two-way ANOVA with Sidak's multiple comparison test, was used to assess between sham and OBx for cutaneous temperature during presentation of EG and 2MT. Figure 3b; D'Agostino-Pearson omnibus test, followed by two-way ANOVA with Sidak's multiple comparison test, was used to assess between ΔD and control for cutaneous temperature during presentation of EG and 2MT. Figure 3c; D'Agostino-Pearson omnibus test, followed by two-way ANOVA with Sidak's multiple comparison test, was used to assess between sham-operated and unilateral TGx mice for cutaneous temperature during presentation of EG and 2MT. Figure 3d; D'Agostino-Pearson omnibus test, followed by two-way ANOVA with Sidak's multiple comparison test, was used to assess between *omp-cre/Trpa1*$^{flox}$ and control mice for cutaneous temperature during with and without presentation of 2MT. Figure 3e; Shapiro–Wilk and Kolmogorov–Smirnov tests, followed by two-way ANOVA with Sidak's multiple comparison test, were used to assess between *adv-cre/Trpa1*$^{flox}$ and control mice for cutaneous temperature during with and without presentation of 2MT. Figure 3f; D'Agostino-Pearson omnibus test, followed by two-way ANOVA with Sidak's multiple comparison test, was used to assess between mice which received intra-TG injection of saline and RTX for cutaneous temperature during with and without presentation of 2MT. Figure 4; we analyzed $n = 5$ for Fig. 4b$_1$, b$_2$, $n = 4$ for Fig. 4b$_3$, c$_1$, c$_2$, d, $n = 3$ for Fig. 4c$_3$, $n = 4$ each for Fig. 4e, f, $n = 5$ each for Fig. 4g–j, and similar results were obtained. Figure 4k; Shapiro–Wilk and Kolmogorov–Smirnov

tests, followed by unpaired two-tailed Student's $t$ test, were used to assess between with and without 4E2MT presentation for total ERK$^+$ cells in the ventral SP5. Figure 4l; Shapiro–Wilk and Kolmogorov–Smirnov tests, followed by unpaired two-tailed Student's $t$ test, were used to assess between with and without 4E2MT presentation for phospho-ERK$^+$ cells in the ventral SP5. Figure 5b; we stained $n = 22$ for *Trpa1-Cre$^+$* and $n = 2$ for *Trpa1-Cre$^-$*, and obtained similar results. Figure 5c; D'Agostino & Pearson test, followed by one-way ANOVA with Dunnett's multiple comparison test, was used to assess between saline IP in *Trpa1-Cre$^+$* mice and each of two other conditions. Figure 5d; Kolmogorov–Smirnov test, followed by unpaired one-tailed Student's $t$ test, was used to assess between *Trpa1-Cre$^+$* and control mice infected with AAV for cutaneous temperature after CNO administration. We examined whether cutaneous temperature would be decreased by CNO administration in the *Trpa1-Cre$^+$* mice compared with that in the control mice. We evaluated the cutaneous temperature reduction using one-tailed test. Figure 5e; D'Agostino & Pearson test, followed by paired one-tailed Student's $t$ test, was used between saline IP and CNO IP for cutaneous temperature in *Trpa1-Cre$^+$* infected with AAV. We examined whether cutaneous temperature decreased by CNO administration. We evaluated the cutaneous temperature reduction using one-tailed test. Figure 5f; Shapiro–Wilk and Kolmogorov–Smirnov tests, followed by Mann–Whitney test, were used between *Trpa1-Cre$^+$* and control mice infected with AAV for cutaneous temperature after saline administration. Figure 6b; we stained $n = 8$ for *Trpa1-Cre$^+$* and $n = 6$ for *Trpa1-Cre$^-$*, and obtained similar results. Figure 6c; Shapiro–Wilk and Kolmogorov–Smirnov tests, followed by Kruskal–Wallis with Dunn's multiple test, were used to assess between *c-fos$^+$* cell numbers for saline IP in *Trpa1-Cre$^+$* and each of two other conditions. Figure 6d; Shapiro–Wilk and Kolmogorov–Smirnov tests, followed by one-tailed Mann–Whitney test, were used between saline and CNO administrations for cutaneous temperature in *Trpa1-Cre$^+$* mice infected with AAV. We examined whether cutaneous temperature decreased by CNO administration. We evaluated the cutaneous temperature reduction using one-tailed test. Figure 6e; Shapiro–Wilk and Kolmogorov–Smirnov tests, followed by one-tailed Wilcoxon matched pairs signed rank test, were used between saline and CNO administrations for cutaneous temperature in control mice infected with AAV. We examined whether cutaneous temperature decreased by CNO administration. We evaluated the cutaneous temperature reduction using one-tailed test. Figure 7b; Kolmogorov–Smirnov tests, followed by Kruskal–Wallis with Dunn's multiple comparisons, were performed between excised-patch currents induced by saline and each of test compounds for excised patch-current. Figure 7c; Kolmogorov–Smirnov test followed by Kruskal–Wallis with uncorrected Dunn's test was performed between calcium responses for all recorded cells in saline condition and those in responsive cells for each of the test compounds. Figure 7d; Kolmogorov–Smirnov test followed by Kruskal–Wallis with uncorrected Dunn's test was performed between calcium responses for all recorded cells in saline condition and those in responsive cells for each of the test compounds. Figure 8a; Shapiro-Wilk test, followed by Kruskal–Wallis with uncorrected Dunn's test, was performed between saline and each of the test compounds for *c-fos* expression. Figure 8b; Shapiro-Wilk test, followed by Kruskal–Wallis with uncorrected Dunn's test, was performed between saline and each of the test compounds for *c-fos* expression. Figure 9; differentially expressed transcripts were identified using likelihood ratio test of Sleuth R package v0.30.0 (ref. [62]) with $q$-value of <0.01. Figure 10a; Kolmogorov–Smirnov tests, followed by one-way ANOVA with Dunnett's multiple comparison test, were used to assess between saline and each of the test compounds for survival time. Figure 10b; Shapiro–Wilk and Kolmogorov–Smirnov tests, followed by unpaired one-tailed Student's $t$ test, were used to assess between *Trpa1$^{-/-}$* and *Trpa1$^{+/-}$* for cutaneous temperature after 4E2MT administration. IP administration of 4E2MT led to a reduction in cutaneous temperature. The suppression of this effect was evaluated using one-tailed test. Figure 10c; Shapiro–Wilk and Kolmogorov–Smirnov tests, followed by one-tailed Mann–Whitney test, were used to assess between *Trpa1$^{-/-}$* and *Trpa1$^{+/-}$* for survival time. IP administration of 4E2MT prolonged survival in 4% oxygen. The suppression of this effect was evaluated using one-tailed test. Figure 10d; log-rank test was performed between 2MT and TMO administration for survival rate. Supplementary Fig. 1; Shapiro–Wilk and Kolmogorov–Smirnov tests, followed by unpaired one-tailed Student's $t$ test, were used to assess between sham and OBx for cutaneous temperature and heart rate in response to 2MT presentation. 2MT presentation led to core temperature reduction. Suppression of this effect was evaluated using one-tailed test. Supplementary Fig. 2; D'Agostino & Pearson test, followed by unpaired one-tailed Student's $t$ test, was used to assess the difference between the ΔD(cng) and control mice. 2MT presentation led to core temperature reduction. Suppression of this effect was evaluated using one-tailed test. Supplementary Fig. 3; Shapiro–Wilk and Kolmogorov–Smirnov tests, followed by Mann–Whitney test, were used to assess between sham and unilateral cervical vagotomy for cutaneous temperature in response to 2MT presentation; Shapiro–Wilk and Kolmogorov–Smirnov tests, followed by unpaired one-tailed Student's $t$ test, were used between sham and bilateral ablation of VG below the diaphragm for cutaneous temperature in response to 2MT presentation. 2MT presentation led to core temperature reduction. Suppression of this effect was evaluated using one-tailed test. Supplementary Fig. 4; D'Agostino & Pearson test, followed by Kruskal–Wallis with Dunn's multiple comparison test, was used to assess between every pairs among three conditions for MnPO, VMPO, and PBN; Shapiro–Wilk and Kolmogorov–Smirnov tests, followed by Kruskal–Wallis with

Dunn's multiple comparison test, were used to assess between every pairs among three conditions for NTS. Supplementary Fig. 5; we analyzed four animals and obtained similar results. Supplementary Fig. 7; we analyzed six sections from three individual animals and obtained similar results. Shapiro–Wilk and Kolmogorov–Smirnov tests, followed by unpaired one-tailed Student's $t$ test, were used to assess the difference between control and lesion sides. 2MT presentation led to an increase in *c-fos* expression in the Sp5. Suppression of this effect was evaluated using one-tailed test. Supplementary Fig. 8; D'Agostino & Pearson test, followed by unpaired one-tailed Student's $t$ test, was used to assess between saline IP and 0.01% CNA IP, Shapiro–Wilk test followed by Mann–Whitney test was used to assess between saline IP and 0.1% CNA IP; Shapiro–Wilk and Kolmogorov–Smirnov tests, followed by unpaired one-tailed Student's $t$ test, were used to assess between saline IP and 1% CNA IP, and between saline IP and 10% CNA IP. We examined whether CNA administration could prolong survival time in 4% oxygen. This effect was evaluated using one-tailed test. Supplementary Fig. 9d; one-way ANOVA with Dunnett's multiple comparisons test was performed to assess between saline and each compound.

**Reporting summary**. Further information on research design is available in the Nature Research Reporting Summary linked to this article.

## Data availability

The RNAseq data are available from https://www.hgvd.genome.med.kyoto-u.ac.jp/repository/MGE0000001.html. The relevant data are available from the corresponding author on reasonable request. Source data are provided with this paper.

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

## Acknowledgements

We are grateful to Fumihiro Eto for providing technical assistance. We are grateful to Drs. Shigetada Nakanishi and Tatsuo Kinashshi for critical comments on the manuscript. This work was supported by the following foundations: JSPS KAKENHI (16K07445 to T.M.; 16H06142 to T.I.; 20H04849, 18H02546, 17H05586, and 16K14558 to R.K.; 20K20578, 18K19350, 18H04806, and 16H02591, to K.K.); the Japan Science and Technology Agency, A-STEP grant to R.K.; the Takeda Science Foundation (to T.M., T.I., R.K., and K.K.); the Canon Foundation (to K.K.); Mishima Kaiun Memorial Foundation (to T.M.); the Dai-ichi Sankyo Foundation (to R.K. and K.K.); the Naito Foundation (to T.I. and K.K.); the Sumitomo Foundation (to K.K.); the Uehara Foundation (to K.K.); the Asahi Glass Foundation (to K.K.); and the Terumo Foundation (to K.K.).

## Author contributions

K.K. designed the study and experiments. K.K. wrote the manuscript with R.K. T.I. performed the experiments shown in Fig. 1 with L.T., T.M., N.K., and S.M. T.M. performed the experiments represented in Fig. 2 with L.T., R.K., and K.K. T.M performed the experiments shown in Fig. 3a–c and Supplementary Fig. 3 with T.I., L.T., A.Y., D.K., R.K., and K.K. T.M. performed the experiment shown in Fig. 3d, e with L.C., C.Y.L., and Q.L. T.M. performed the experiments shown in Fig. 4 with D.K., M.I., R.K., and K.K. T.M performed the experiments depicted in Figs. 5, 6, and Supplementary Fig. S3–S5 with T.I., L.T., A.Y., D.K., R.K., and K.K. A.D. performed the experiments represented in Fig. 7b with M.H. Y.H. performed the experiment shown in Fig. 7c, d and Supplementary Fig. 6. T.M. and T.I. performed the experiments presented in Fig. 8a, b with R.K. and K.K. T.M. performed the experiments presented in Fig. 9 with K.H., R.K., and K.K. T.M. performed the experiments shown in Fig. 10 with L.T., R.K., and K.K. T.M. performed the experiments presented in Supplementary Fig. 1 with K.K. T.M. performed the experiments presented in Supplementary Fig. 2, with L.T. and R.K. T.I. performed the experiments shown in Supplementary Fig. 4 with A.Y., R.K., and K.K. T.I. performed the experiment shown in Supplementary Fig. 5 with D.K. K.K. performed the experiment shown in Supplementary Fig. 7 with T.M. and R.K. K.K. performed the experiment shown in Supplementary Fig. 8 with R.K. K.K. performed the experiment shown in Supplementary Fig. 9 with T.M., A.Y., and R.K. T.M. performed the experiments presented in Supplementary Fig. 10 with T.M., I.Y., and M.S.

## Competing interests

The authors declare no competing interests.
