## [Peer Review File · Nature Communications]

Reviewers' Comments:

Reviewer #1:

Remarks to the Author:

Animals must navigate a diverse chemosensory world. While odors and tastes are perhaps most familiar, compounds with chemesthetic activity are also quite common. These compounds, many of which are found in vegetation such as herbs, spices and chilis, exert their effects by stimulating trigeminal and vagal nerve endings, often via activation of ion channels such as members of the TRP family. Chemesthetic responses are often sentinel in their function, warning the animal of potential danger and promoting protective responses such as inflammation or avoidance.

In this paper, the authors describe a large set up studies that show members of the thiazolines at via Trpa1-containing cranial nerves to promote protective responses in mice, in particular hypothermia. The authors present a thorough set of studies that support their primary conclusions, and some of the results (such as the ability of TMO to protect during hypoxia) are quite striking. However, a few concerns should be addressed.

1) The authors' backgrounds as olfactory biologists has perhaps biased the way that they consider the chemical stimuli with which they are working here. It is true that several of the compounds they use do activate the olfactory system and thus can function as odorants. However here it is clear these chemical stimuli are acting as chemesthetic agents, not odorants, and should not be called such. It would also benefit the reader if the authors spent some more time in the Discussion to put their results into context of the extensive chemesthesis literature.

2) Many times the authors go beyond putting their study in the context of potential therapeutics to talk extensively about the implications of their findings for treatments of "life-threatening conditions." Some discussion of these implications in the Discussion is fine, but the connection between the results here in mice and any therapeutic outcomes in humans is far too tenuous at this point. The study as it is is plenty interesting and does not need to be oversold. Linking the results to potential therapies should be toned down throughout, and removed from the abstract.

3) The differential activity of the Trpa1 agonist cinnamaldehyde when it came to activation of the spinal nucleus was both puzzling and intriguing. However, the transcriptomic analysis seems to be much effort for little payoff. We are not left with any explanation of how multiple agonists of the same receptor channel could exert such different effects in trigeminal neurons, nor how this would impact more central activation. Unconsidered, it seems, is the possibility that compounds with differential impacts of activity in the CNS may be activating other sensory neurons that modify this activity somewhere along the neuraxis. This concept is already clear in these experiments as many of these compounds activate both the olfactory system and the trigeminal/vagal nerves. This needs to be discussed more thoroughly.

Reviewer #2:

Remarks to the Author:

The manuscript by Matsuo and colleagues investigate the mechanisms by which predator-related odors (tFOs) induce protective physiological responses in mice. They rely on their previous findings from a bioRxiv preprint and on a diverse set of experimental approaches to show that tFOs activate Trpa1 channels in trigeminal and vagal neurons, and that this leads to activation of spinal trigeminal tract and of nucleus tractus solitarius, and subsequently to hypothermia. Other protective effects like bradycardia, decreased oxygen consumption and anti-inflammatory responses seem to also require Trpa1.

The authors also go on to test known and novel ligands for Trpa1, in search for potential therapeutic molecules.

Overall, I believe that this a good and important paper, with great translational potential, but I have certain concerns. Some are major and some are minor, I list them in the order they appear in text:

Fig. 1F – it appears that heart rate dropped more in *Trpa1* ^{-/-} than in control mice – in the text that is not mentioned. What would be the explanation for it?

Fig. 1E,F – what do the different shaded areas represent?

Page 10 – citation '30' is added in reference to 'respectively'?

Figure 3A – here the authors use the abbreviation 'NTS' which is the conventional one, but everywhere else in the manuscript they seem to use 'NST'

Figure 3I,L,M – in general this figure seems to be descriptive, but some controls for c-fos expression seem needed. For example how does c-fos expression in Sp5 and NST look after non-predatory odors? I realize that this is in a way addressed in Figure 5D, but is lacking at this point in the paper.

Figure 3N-Q – what is the expression pattern of total ERK ? shouldn't that be shown as control?

Figure S2 – there is no clear explanation/discussion in the text as to why sub-diaphragmic vagotomy but not cervical vagotomy has an effect

Is there statistical evidence for this claim: 'Among these three sensory pathways, the trigeminal pathway appeared to contribute most to the regulation of 2MT-induced hypothermia.'

Figure 4G: if I understand the experiment correctly, c-fos could be activated by non-TG and non-VG *Trpa1*+ neurons projecting to Sp5.

Figure 4I and J: shouldn't there be a 'saline' control experiment in *Cre*+ mice?

Figure 4L: why is the cutaneous temperature data here presented as difference (delta), but in degrees C in previous figures? How does the 'rescue' temperature compares to the 2MT-induced hypothermia in wild-type mice? Is this a full rescue, a partial rescue?

The citation for this statement is missing: 'We previously showed that both vaporized odor stimulation and IP injection of tFOs upregulated c-fos expression in the Sp5/NST and also induced bioprotective effects'

What would be the mechanism for the effects of i.p. injections? Similar? More focused on VG?

Similarly, what is the mechanisms for anti-inflammatory effects?

I wonder if, instead of adding the anti-inflammatory data, the authors could go in more depth with the mechanisms for hypothermia (for which they have more data), and link Sp5 and NST activation with activity in known thermoregulatory centers (MnPO, VMPO, etc).

Reviewer #3:

Remarks to the Author:

Matsuo and colleagues report an interesting series of experiments that explore the role of transient receptor potential ankyrin type1 (TRPA1) channels in supposedly bioprotective responses to danger situations. These responses include hypothermia, reduction of oxygen consumption and anti-

inflammatory effects. The authors claim that thiazoline-related chemicals, which are said to be potent fear-eliciting odorants (e.g. predator kairomones such as TMT in fox feces), are indeed ligands of TRPA1 receptors and this would be the basis of their supposed ability to induce other bioprotective responses related to altered physiology and homeostasis.

There are very interesting findings in the manuscript. But also, some conceptual and methodological issues that require being worked out.

THE CONCEPT OF ODORANT

Already in the title, and throughout the manuscript, the authors use the term innate fear odours. I don't like this expression; I would prefer fear-eliciting odours. But the true question is if tFOs are actually odorants. An odorant is a substance, more or less volatile, detected by the olfactory epithelium and giving rise to the perception of odour. One of the outcomes of the work is that tFOs are TRPA1 ligands activating some vagal and trigeminal sensory ganglion cells and, as a consequence, a central circuit that has the first relay in portions of the Sp5 and NST.

The main conclusion that can be drawn from these findings is that tFOs constitute trigeminal/vagal somatosensory (probably nociceptive) stimuli, not odorants. Even if tFOs can have an odour, in fact a strong one to the human nose, it cannot be said that the effects that the authors report are due to their odorous nature. In fact, in most experiments, tFOs are intraperitoneally injected (page 38: 100 μ L of a 1% 'odor' solution, in saline? In water? In DMSO?) to get the effects the authors investigate. Methodological details of each experiment are needed to clarify this issue but, the structure of the papers published in Nature Communications and other highly cited journals is not especially useful in that respect.

For one of the experiments, the authors also report exposure to odorants (2MT; four set of filter paper scented with 271 μ mol of the odorant; each one?) in a semi-closed cage (17.5x10.5x15 cm). In these conditions, animals show hypothermia (both central and cutaneous) and bradycardia (Fig. 1). This experiment does not demonstrate that 2MT is acting as an odorant, just the opposite. Its effects on heart rate and body temperature are mediated by its binding to trpa1 (no bradycardia and hypothermia is induced in trpa1^{-/-} mice), rather than to its binding to olfactory receptors. A substance having an odour, is not always used as an odorant. For instance, chloral hydrate has an odour, but when it is used as a sedative, nobody would name it as an odorant. It is not its odour what causes sedation. In the same way, it is not tFOs odour what causes bradycardia and hypothermia.

This experiment includes analyses of bradycardia and hypothermia induced by a "restrained condition". The authors must describe these "restrained conditions", there is no reference to this experiment in the Methods section.

When high concentrations of an odorant are used, behavioural reactions are said to be contaminated by trigeminal stimulation. This seems to happen with tFOs (Galiano et al., 2012; lesion of the olfactory epithelium does not preclude TMT-induced immobility) and even with food-related odorants such as isoamyl-acetate, which is not only avoided (Wallace and Rosen, 2000) but even elicit fear or immobility in rats (Wallace and Rosen, 2000; experiment 1 in Rosen et al., 2006; see discussion in Fortes-Marco et al., 2013). The nose and respiratory tract are innervated by trigeminal and vagal sensory fibres and highly concentrated trigeminal stimuli (such as tFOs) are surely giving rise to nociception that might explain the effects observed. If you have 'tasted' wasabi, this is easy to understand.

There is a simple way to test this. It can be analysed if animals exposed to 2MT "in a semi-closed cage" show activation of trigeminal/vagal ganglion cells and in the Sp5/NTS/AP region.

As the authors conclude (page 11-12), "Trpa1 in the trigeminal nerves regulates tFO-induced hypothermia". Although olfactory bulb ablation reduces 2MT-induced hypothermia, the authors further check if this effect might be due to altered trigeminal innervation of the nasal region. To do so they check if AD mutant mice, in which the dorsal nasal cavity is depleted of olfactory sensory neurons

(some of them sensitive to TMT; Kowayakawa et al., 2007), show 2MT-induced hypothermia: there is a reduction of hypothermia in AD mice (Fig. 4B). However, depletion of *trpa1* in olfactory sensory neurons has no effect on 2MT-hypothermia. From these data, it seems that AD mice might have not just depletion of olfactory neurons in the dorsal olfactory epithelium but also altered trigeminal innervation of the nasal cavity.

In summary, altogether the findings of the authors in such an impressive array of complex experiments suggests that trigeminal rather than olfactory innervation is responsible of the observed effects of outer exposure to tFOs in a semi-closed cage. Therefore, using the term odorants to name these substances in the current context is confusing and misleading for the reader. I would change the title and the rest of the text to clarify this issue.

REPRODUCIBILITY and STATISTICS

One of the issues that worries current science panorama is the lack of reproducibility. This is one of the reasons why methodology must be described in high detail. This is also one of the reasons why I don't like the use of lots of supplementary material, very often with few details on procedure.

Figure 2A shows that 2MT induces an increase in oxygen consumption in *trpa1*^{-/-} mice, but a decrease in *trpa1*^{+/-} ones. Consequently, survival in severe hypoxic conditions (4% oxygen) is very short in *trpa1*^{-/-} as compared to *trpa1*^{+/-} animals (Figure 2B). Figure 2D shows an additional experiment in which saline and acetaminophen (APAP) are administered to *trpa1*^{-/-} and *trpa1*^{+/-} mice and their survival in hypoxic conditions tested. APAP, a ligand of TRPA1, increases survival in control but not *trpa1*^{-/-} animals.

There is a lack of methodological details in the description of the experiment (in general in the experiments). Is 2MT exposure performed like in the experiment reported in Fig. 1 (soaking filter paper and putting 4 papers in a semi-closed cage)? Or like in Figures 3 and 7 (i.p injection)? Statistics is also an important issue here. According to the figure legend, Student's t-tests are used. Many of the experiments reported require a two-way ANOVA, with genotype as one of the variables and treatment as the second one, instead of simple t tests (Fig. 1; Fig. 2D; Fig. 4; Fig. 5C and 5D). In other cases (e.g. Fig. 7A or 5B) and ANOVA followed by Dunnett's post-hoc (comparing each treatment with control), instead of simple (iterative) Student's t tests, or Bonferroni if comparisons between different treatments are intended. Experiment reported in figure 7E probably requires a Kaplan-Meier survival test.

There is no mention of normality and homoscedasticity requirements in any figure (required for t tests), and it is hard to believe that all the data fulfil these requirements in all experiments. In fact, there is subsection on statistics (just a simple paragraph) in the section of Methods.

FIGURE 3 and brain activation by IP-injected tFOs

Figure 3 shows morphological details of the innervation of vagal and trigeminal centres by *trpa1*-expressing cells, and also of activity-related markers (p-ERK and c-fos mRNA) in response to IP injection of 4E2MT. Immunoreactivity for p-ERK and c-fos mRNA should be compared between 4E2MT- and saline-exposed animals. Otherwise we ignore if this labelling is due to 4E2MT or it is constitutive. Although the involvement of the vagal/trigeminal pathways on the biological effects of these drugs is likely, there is no clear demonstration of it in the results of the manuscript as currently reported.

BIOPROTECTIVE EFFECTS OF tFOs

I really don't grasp the use of the term "bioprotective" in the context of the experiments of the

manuscript. Concerning anti-inflammatory actions of some tFOs, it is reported to reduce TNF-alpha in response to LPS injection. This is, per se, not bioprotective, as inflammatory responses are bioprotective or adaptive in some circumstances. Concerning the hypothermia induced by tFOs, it is bioprotective if there is fever or hyperthermia. Finally, survival under severe hypoxia is enhanced, and therefore this is a bioprotective effect in strange circumstances in which there is hypoxia. For instance, if there is stroke there cell survival under local hypoxia would arguably be bioprotective. But the trigeminal/vagal-mediated effects of tFOs are not possibly local.

I can't understand the relationship of H₂S-inhibition of mitochondrial respiration and the affair of tFOs. The paragraph of the Introduction on that issue is difficult to understand and very confusing: "Low concentration of H₂S is thought to partially inhibit mitochondrial respiratory activity [22], thereby enhancing the ability to survive in a hypoxic environment". I cannot understand this statement that looks, at least, counter-intuitive.

CONCLUSION

The manuscript by Matsuo and collaborators reports a series of interesting experiments on a putatively promising issue (ligands of trpa1 receptors, and their biological effects). But it has important methodological problems related to statistics and experimental design (c-fos and p-ERK). In addition, there are also conceptual issues that need the attention of the authors: the repeated use of "odorant" to refer to trpa1-ligands, when their results indicate that they likely are vagal/trigeminal stimuli rather than olfactory ones. And the use of the expression "bioprotective effects" to refer to simple anti-inflammatory and hypothermic effects apparently mediated by (according to authors results) the effect of these drugs on peripheral fibres of the somatosensory system. The relationship between antipredatory (fear) reactions and this stimulation of vagal/trigeminal fibres is very unlikely, so that speaking of tFOs and mediating survival mechanisms in life-threatening situations seems not appropriate.

Point-by-point Response to Reviewers' Comments

We are grateful to the reviewers for their valuable comments. These comments have been very constructive and helped us to improve the scientific value of our manuscript, specifically in terms of clarifying our terminology and the goal of this paper. We address (in black font) each of the reviewers' concerns (in blue font) in this letter in an itemized fashion, and corresponding changes have been made to improve the manuscript.

We look forward to the outcome of your assessment. Thank you.

Reviewer #1 (Remarks to the Author)

Animals must navigate a diverse chemosensory world. While odors and tastes are perhaps most familiar, compounds with chemesthetic activity are also quite common. These compounds, many of which are found in vegetation such as herbs, spices and chilis, exert their effects by stimulating trigeminal and vagal nerve endings, often via activation of ion channels such as members of the TRP family. Chemesthetic responses are often sentinel in their function, warning the animal of potential danger and promoting protective responses such as inflammation or avoidance.

In this paper, the authors describe a large set up studies that show members of the thiazolines at via Trpa1-containing cranial nerves to promote protective responses in mice, in particular hypothermia. The authors present a thorough set of studies that support their primary conclusions, and some of the results (such as the ability of TMO to protect during hypoxia) are quite striking. However, a few concerns should be addressed.

1) The authors' backgrounds as olfactory biologists has perhaps biased the way that they consider the chemical stimuli with which they are working here. It is true that several of the compounds they use do activate the olfactory system and thus can function as odorants. However here it is clear these chemical stimuli are acting as chemesthetic agents, not odorants, and should not be called such. It would also benefit the reader if the authors spent some more time in the Discussion to put their results into context of the extensive chemesthesis literature.

Response: We thank the reviewer for raising this critical point.

As pointed out, heterocyclic compounds such as 2MT activate the trigeminal and vagus nerves in addition to the olfactory receptors. Thus, these molecules act as both odorant and chemesthetic compounds; the term "odorant" gives the strong impression that they are eliciting responses via the olfactory pathway, underestimating the chemesthetic pathway's contribution. To solve this problem, we refrained from using "odorants" in describing the compounds in this study. Instead, we either used the specific compound names, e.g., 2-methyl-2-thiazoline (2MT) or used the terms "compound" and "molecules" in the revised manuscript.

We have also fine-tuned the terminology, from "odor presentation" and "no odor condition" to "presentation of test compound" and "control condition" in the revised manuscript (indicated by cyan highlights). Likewise, we have used the term "tFO" as a generic term for active heterocyclic compounds in sections with word limits, such as the Summary, and when multiple compounds are simultaneously described. In this case, to clarify that "tFO" does not refer only to the odorant, we have defined tFO as a thiazoline-related innate fear-eliciting compound in lines 34 and 60. As pointed out by Reviewer #1, it is important that a single compound has both

odorous and chemesthetic properties, which is directly linked to the conclusion of this study. Therefore, we have added a new section in the Discussion for this point in the revised manuscript (lines 375-383).

2) Many times the authors go beyond putting their study in the context of potential therapeutics to talk extensively about the implications of their findings for treatments of "life-threatening conditions." Some discussion of these implications in the Discussion is fine, but the connection between the results here in mice and any therapeutic outcomes in humans is far too tenuous at this point. The study as it is is plenty interesting and does not need to be oversold. Linking the results to potential therapies should be toned down throughout, and removed from the abstract.

Response: As pointed by Reviewer# 1, this study's potential medical applications are unknown at present. Therefore, sentences referring to our findings' possible clinical implications have been deleted from the Summary and Introduction (lines 47, 98).

3) The differential activity of the Trpa1 agonist cinnamaldehyde when it came to activation of the spinal nucleus was both puzzling and intriguing. However, the transcriptomic analysis seems to be much effort for little payoff. We are not left with any explanation of how multiple agonists of the same receptor channel could exert such different effects in trigeminal neurons, nor how this would impact more central activation. Unconsidered, it seems, is the possibility that compounds with differential impacts of activity in the CNS may be activating other sensory neurons that modify this activity somewhere along the neuraxis. This concept is already clear in these experiments as many of these compounds activate both the olfactory system and the trigeminal/vagal nerves. This needs to be discussed more thoroughly.

Response: As pointed by Reviewer #1, cinnamaldehyde (CNA), a well-known TRPA1 agonist, did not activate Sp5/NTS unlike tFOs (such as 2MT and 4E2MT). This study argues for a model that this difference may have an important relation with anti-hypoxic activity. However, because the mechanisms of how such differences are regulated and how they impact central activation are unknown, various possibilities need to be considered.

Reviewer #1 pointed out that Sp5/NTS activation may be regulated by integrated inputs from sensory neurons other than the trigeminal and vagus nerves; because tFO activates the olfactory neuron in addition to the trigeminal and vagus nerves, the possible contribution of the olfactory neurons needs to be verified. Therefore, we conducted a new experiment to analyze CNA- and tFOs-induced *c-fos* mRNA expression in the olfactory bulb.

Different compounds clearly induced different *c-fos* mRNA expression levels in the olfactory bulb. However, there was no clear correlation between the *c-fos* mRNA expression in the olfactory bulb and the induction of *c-fos* mRNA expression in SP5 and anti-hypoxic activity. These results suggest that the differences in olfactory bulb input may not regulate the activation of Sp5 or anti-hypoxic activity. We have added this new experiment's results and discussed the different TRPA1 agonists with their differential effects on Sp5/NTS activation and hypoxic resistance (lines 337-348).

Reviewer #2 (Remarks to the Author)

The manuscript by Matsuo and colleagues investigate the mechanisms by which predator-related odors (tFOs) induce protective physiological responses in mice. They rely on their previous findings from a bioRxiv preprint and on a diverse set of experimental approaches to show that tFOs activate *Trpa1* channels in trigeminal and vagal neurons, and that this leads to activation of spinal trigeminal tract and of nucleus tractus solitarius, and subsequently to hypothermia. Other protective effects like bradycardia, decreased oxygen consumption and anti-inflammatory responses seem to also require *Trpa1*.

The authors also go on to test known and novel ligands for *Trpa1*, in search for potential therapeutic molecules.

Overall, I believe that this a good and important paper, with great translational potential, but I have certain concerns. Some are major and some are minor, I list them in the order they appear in text:

Response: We thank the reviewer for raising numerous important points. In response to the reviewer's feedback, we have conducted additional experiments and revised the manuscript accordingly. We hope the modifications will result in a manuscript that more clearly and accurately conveys the points to be argued in this paper.

The initial manuscript emphasized that the TG pathway plays the most crucial role in the tFO-induced physiological responses. However, based on suggestions raised by Reviewer #2, we reviewed our previous data and performed further experiments, as shown in the revised Figure 4 M-Q and Figure S3. These new experiments more clearly demonstrated the important role of the VG pathway (Figure 4M-Q) and the olfactory system's contribution (Figure S3) in regulating tFO-induced physiological responses. By revising the manuscript with these additional experiments, we believe we have more comprehensively depicted tFOs as odorant and chemesthetic agent, which induces physiological responses via multiple sensory systems.

Also in response to the reviewer's advice, we have also removed the anti-inflammatory effect results, adding the new experiments' results on the relation of brain regions involved in tFO-induced hypothermia with known hypothermia-controlling brain regions instead. The results suggest that tFO-induced hypothermia may be regulated by a brainstem to midbrain pathway, different from the known hypothermia control system controlled by the hypothalamus. We believe this improvement focuses on the mechanism behind tFO-induced hypothermia, which clarifies the argument of the paper.

Fig. 1F – it appears that heart rate dropped more in *Trpa1* ^{-/-} than in control mice – in the text that is not mentioned. What would be the explanation for it?

Response: In this study, we showed physiological responses to restraint stress to argue that hypothermia induced by innate fear stimuli other than 2MT is still observed in *Trpa1* knockout mice. Reviewer# 2 noted that the restraint-induced hypothermia is more potent in *Trpa1* knockout mice than in control mice. As claimed in this paper, *Trpa1* is involved in the initiation of hypothermia induced by innate fear stimuli. In addition, *Trpa1* may also regulate the temperature fluctuation patterns caused by innate fear stimuli; it may have a function in sensing excessive drops in body temperature, hence, playing a role in thermoregulation. *Trpa1* has been reported to sense low temperatures and may be involved in such responses. We have mentioned these possibilities in the Results section (lines 117-122).

Fig. 1E,F – what do the different shaded areas represent?

Response: We compared cutaneous temperature and heart rate after 2MT presentation (11-20 min) between *Trpa1* knockout and control mice (right bar graphs). This duration was marked by the shaded areas. We have clarified its description in the figure legend of Figure 1 (lines 986 and 988).

Page 10 – citation ‘30’ is added in reference to ‘respectively’?

Response: Thank you for pointing this out. We have corrected the placement of the citation in the revised manuscript (line 170).

Figure 3A – here the authors use the abbreviation ‘NTS’ which is the conventional one, but everywhere else in the manuscript they seem to use ‘NST’

Response: As advised, “NTS” has been used throughout the revised manuscript in place of “NST.”

Figure 3L,M – in general this figure seems to be descriptive, but some controls for *c-fos* expression seem needed. For example how does *c-fos* expression in Sp5 and NST look after non-predatory odors? I realize that this is in a way addressed in Figure 5D, but is lacking at this point in the paper.

Response: We have added the data for *c-fos* mRNA expression in the Sp5 and NTS in control conditions to show *c-fos*’ basal expression in these areas (Figure 3E).

Figure 3N-Q – what is the expression pattern of total ERK ? shouldn’t that be shown as control?

Response: In response to this comment, we have performed immunohistochemical studies to detect the total ERK in response to tFO stimulation as a control (Figure 3G-K).

Figure S2 – there is no clear explanation/discussion in the text as to why sub-diaphragmatic vagotomy but not cervical vagotomy has an effect

Response: We found that performing bilateral cervical vagotomy caused the mice to die quickly. Therefore, a unilateral cervical vagotomy was performed. However, the latter procedure did not affect 2MT-induced hypothermia, leading us to perform bilateral sub-diaphragmatic vagotomy instead. As mentioned in the manuscript, 2MT-induced hypothermia was suppressed in this set-up.

In the revised manuscript, we have more clearly stated that cervical vagotomy was performed unilaterally, whereas sub-diaphragmatic vagotomy was performed bilaterally (lines 211-220; revised Figure S5).

Is there statistical evidence for this claim: ‘Among these three sensory pathways, the trigeminal pathway appeared to contribute most to the regulation of 2MT-induced hypothermia.’

Response: In this study, we showed that the olfactory, trigeminal, and vagus nerves are involved in tFO-induced hypothermia and hypoxia resistance. The experimental methods for analyzing these three sensory pathways' functions are different; therefore, it was not possible to quantitatively compare the three neural pathways' contributions. In addition, we have conducted additional experiments showing the contributions of the olfactory and *Trpa1*-positive vagal systems to clearly show the role of non-TG pathways (Figures S3, S4, and 4M-Q). Thus, in the revised manuscript, we have refrained from stating that the trigeminal pathway makes the largest contribution. In its place is the statement that three neural pathways—specifically the olfactory, trigeminal, and vagal—are involved in tFO-induced physiological responses (lines 220-221).

Figure 4G: if I understand the experiment correctly, c-fos could be activated by non-TG and non-VG *Trpa1*+ neurons projecting to Sp5.

Response: We injected a Cre-dependent retrograde AAV encoding hM3Dq into the dorsal and ventral SP5 (Sp5d and Sp5v) of *Trpa1-Cre* mice. Therefore, as pointed by Reviewer #2, Gq-DREADD is likely to be expressed in Cre-positive neurons projecting to the Sp5d and Sp5v and not just TG. Fig 4G has been modified to clearly show this point.

Figure 4I and J: shouldn't there be a 'saline' control experiment in Cre+ mice?

Response: In response to the comment, we have performed additional experiments for saline control in *Trpa1-Cre*⁺ mice (Figures 4I, K, L).

Figure 4L: why is the cutaneous temperature data here presented as difference (delta), but in degrees C in previous figures? How does the 'rescue' temperature compares to the 2MT-induced hypothermia in wild-type mice? Is this a full rescue, a partial rescue?

Response: In the experiments other than that depicted in the original Figure 4L, we measured the control and test animals' body temperatures on the same day or at least within several days of each other. However, in the experiment shown in the original Figure 4L, the rescued and control animals' body temperatures were measured at very different times, which resulted in disparate baseline temperatures. To solve this problem, the changes in body temperature were shown graphically, unlike in other experiments.

However, when we recalculated the absolute temperature comparisons to address Reviewer #2's point, there was no statistically significant difference between the control and rescue animals. For this reason, we decided to exclude the results of the experiment in the original Figure 4L in the revised manuscript. In its place, we performed additional experiments (shown in Figure 4M-Q in the revised manuscript), which showed that not only *Trpa1*-positive neurons projecting to the SP5, including TG, are involved in the regulation of tFO-induced hypothermia, but also other such neurons projecting to the NTS, including VG.

The citation for this statement is missing: 'We previously showed that both vaporized odor stimulation and IP injection of tFOs upregulated c-fos expression in the Sp5/NST and also induced bioprotective effects '

Response: Thank you for pointing this out. We have added a corresponding citation in the revised manuscript (line 292).

What would be the mechanism for the effects of i.p. injections? Similar? More focused on VG?

Response: It is widely accepted that both vaporization and IP injection of odorants activate sensory perception. Accordingly, we checked the *c-fos* mRNA expression induced by IP injection of tFO and found that its expression was observed in both SP5 and NTS, similar to the vaporized presentation of 2MT. These results are shown in bioRxiv (Figure S4 in reference 5). We have added this notion in the revised manuscript with references to show the rationale of IP injections of tFOs (lines 289-293, and references 5, 47, 48).

Similarly, what is the mechanisms for anti-inflammatory effects?

I wonder if, instead of adding the anti-inflammatory data, the authors could go in more depth with the mechanisms for hypothermia (for which they have more data), and link Sp5 and NST activation with activity in known thermoregulatory centers (MnPO, VMPO, etc).

Response: This study does not present a detailed analysis of tFO-induced inflammatory effects. We agree with Reviewer #2 suggestion of focusing on tFO-induced hypothermia, which will also provide a more linear story for readers. Thus, as advised, we have excluded the anti-inflammation data and conducted additional experiments to show the relevance of tFO-induced hypothermia and known hypothermia-inducing pathways.

Previous studies showed that the MnPO and VMPO play important roles in regulating hypothermia (references 27-30). On the other hand, we showed that the NTS-PBN pathway has a significant role in regulating tFO-induced hypothermia (ref 5) and performed additional *c-fos* mapping experiments concerning the known hypothermia control centers (e.g. MnPO, VMPO) and NTS-PBN pathways.

Concerning the MnPO and VMPO, tFO-stimulation significantly increased *c-fos* mRNA expression only in the VMPO. However, the increase in *c-fos* mRNA expression was still observed in *Trpa1* knockout mice. On the other hand, for the NTS-PBN pathway, tFO-stimulation increased *c-fos* mRNA expression in both the NTS and PBN, with these increases almost completely abolished in *Trpa1* knockout mice. These results suggest that tFO-induced hypothermia mediated by *Trpa1* is regulated by the NTS-PBN pathway rather than the established MnPO and VMPO pathways (Figure S1).

We have previously shown that tFO-induced hypothermia accelerates glucose uptake into the brain and decreases blood oxygen saturation (reference 5). In contrast, in hibernation, the former is reduced and the latter is unchanged. Thus, although known hypothermia-relevant phenomena, e.g., hibernation and torpor, and tFO both decrease body temperature, they are accompanied by different physiological responses. Reflecting this, it is presumed that different neural pathways regulate these hypothermic responses. We have added these descriptions in the revised manuscript (lines 150-166).

Reviewer #3 (Remarks to the Author)

Matsuo and colleagues report an interesting series of experiments that explore the role of transient receptor potential ankyrin type 1 (TRPA1) channels in supposedly bioprotective responses to danger situations. These responses include hypothermia, reduction of oxygen consumption and anti-inflammatory effects. The authors claim that thiazoline-related chemicals, which are said to be potent fear-eliciting odorants (e.g. predator kairomones such as TMT in fox feces), are indeed ligands of TRPA1 receptors and this would be the basis of their supposed ability to induce other bioprotective responses related to altered physiology and homeostasis.

There are very interesting findings in the manuscript. But also, some conceptual and methodological issues that require being worked out.

THE CONCEPT OF ODORANT

Already in the title, and throughout the manuscript, the authors use the term innate fear odours. I don't like this expression; I would prefer fear-eliciting odours. But the true question is if tFOs are actually odorants. An odorant is a substance, more or less volatile, detected by the olfactory epithelium and giving rise to odour perception. One of the outcomes of the work is that tFOs are TRPA1 ligands activating some vagal and trigeminal sensory ganglion cells and, as a consequence, a central circuit that has the first relay in portions of the Sp5 and NST. The main conclusion that can be drawn from these findings is that tFOs constitute trigeminal/vagal somatosensory (probably nociceptive) stimuli, not odorants. Even if tFOs can have an odour, in fact a strong one to the human nose, it cannot be said that the effects that the authors report are due to their odorous nature. In fact, in most experiments, tFOs are intraperitoneally injected (page 38: 100 uL of a 1% 'odor' solution, in saline? In water? In DMSO?) to get the effects the authors investigate. Methodological details of each experiment are needed to clarify this issue but, the structure of the papers published in Nature Communications and other highly cited journals is not especially useful in that respect.

Response: We thank the reviewer for the criticism of this important point. Accordingly, we have revised the manuscript's wording to eliminate the misunderstanding caused by using the term "odors." As Reviewer #3 pointed out, the term "odors" is used for molecules that activate the olfactory pathway, giving rise to odor perception. Previous studies showed that the thiazoline-related molecules used in this study induce fear-related behaviors via the olfactory neurons. Therefore, thiazoline-related molecules, e.g., 2MT and TMT, have been described as odor molecules (references 6-8). However, this study argues that the physiological responses induced by 2MT and its related compounds are controlled via the trigeminal and vagal pathways. Thus, as Reviewer #1 also pointed out, these molecules should be described as chemesthetic agents rather than odors.

We also agree that it is not appropriate to refer to 2MT as an odor, as used in the original manuscript. Thus, we have stopped using the term "odors" in the revised manuscript. Instead, we replaced the wording by either 1) using the name of compound, 2) referring to "compound" or 3) referring to "tFO" when we would like to mention a common feature of multiple thiazoline-related compounds. We have also clearly defined tFO as "thiazoline-related innate fear-eliciting compound" (lines 34 and 60).

For one of the experiments, the authors also report exposure to odorants (2MT; four set of filter paper scented with 271 umol of the odorant; each one?) in a semi-closed cage (17.5x10.5x15 cm). In these conditions, animals show hypothermia (both central and cutaneous) and

bradycardia (Fig. 1). This experiment does not demonstrate that 2MT is acting as an odorant, just the opposite. Its effects on heart rate and body temperature are mediated by its binding to *trpa1* (no bradycardia and hypothermia is induced in *trpa1*^{-/-} mice), rather than to its binding to olfactory receptors. A substance having an odour, is not always used as an odorant. For instance, chloral hydrate has an odour, but when it is used as a sedative, nobody would name it as an odorant. It is not its odour what causes sedation. In the same way, it is not tFOs odour what causes bradycardia and hypothermia.

This experiment includes analyses of bradycardia and hypothermia induced by a “restrained condition”. The authors must describe these “restrained conditions”, there is no reference to this experiment in the Methods section.

Response: We apologize for the confusion caused by the inadequate description of the experimental methods. As pointed out by Reviewer #3, the inadequate description of the experiments prevented the reader from getting accurate information. There was also a problem of forgetting to delete irrelevant sentences at the paper's proofreading stage. Presentation of tFOs by a restrained condition, mentioned by Reviewer#3, was not conducted in this study; therefore, that section has been deleted in the revised manuscript.

Two methods of thiazoline-related compound-stimulation were used in this study. The first condition was the filter paper presentation with 271 mol of a thiazoline-related compound to allow vaporization. In the second condition, 100 ul of thiazoline-related compounds diluted to 1% in saline was injected intraperitoneally. Both conditions were performed in an open cage placed in a chemical fume hood to prevent the volatile molecules' diffusion. These conditions have been specified in the Methods section in the revised manuscript (lines 436-442, 447-450, 462-464, 478-482, 485-487, 489-490, 492-493, 501-516, 525-527, and 636-637). We have also modified the figure legends to clearly indicate whether the experiments shown in each figure were performed under presentation or IP stimulation conditions.

When high concentrations of an odorant are used, behavioural reactions are said to be contaminated by trigeminal stimulation. This seems to happen with tFOs (Galiano et al., 2012; lesion of the olfactory epithelium does not preclude TMT-induced immobility) and even with food-related odorants such as isoamyl-acetate, which is not only avoided (Wallace and Rosen, 2000) but even elicit fear or immobility in rats (Wallace and Rosen, 2000; experiment 1 in Rosen et al., 2006; see discussion in Fortes-Marco et al., 2013). The nose and respiratory tract are innervated by trigeminal and vagal sensory fibres and highly concentrated trigeminal stimuli (such as tFOs) are surely giving rise to nociception that might explain the effects observed. If you have ‘tasted’ wasabi, this is easy to understand. There is a simple way to test this. It can be analysed if animals exposed to 2MT “in a semi-closed cage” show activation of trigeminal/vagal ganglion cells and in the Sp5/NTS/AP region.

Response: Statements that we made were more ambiguous than intended, because our initial manuscript did not describe the exact stimulation method using thiazoline-related compounds. Experiments using vaporized thiazoline-related compounds were not conducted in a semi-closed environment but in an open cage with a stainless steel wire lid. We have adjusted the text in the Methods section of the revised manuscript to be clearer (lines 447-448, 461-463, 466-468, 485-487, 501-503, 507-508, and 512-513).

Specifically, *c-fos* mapping was performed in an open cage in a chemical fume hood by presenting filter papers dropped with 271 umol of 2MT every 5 min for 30 min. In this

condition, *c-fos* mRNA expression was observed in the NST and SP5 (Figure S4A-C in Reference 5). In the present study, we performed *c-fos* mapping using *Trpa1* mice under the same conditions and confirmed *c-fos* expression in the NTS (Figure S1).

As the authors conclude (page 11-12), “*Trpa1* in the trigeminal nerves regulates tFO-induced hypothermia”. Although olfactory bulb ablation reduces 2MT-induced hypothermia, the authors further check if this effect might be due to altered trigeminal innervation of the nasal region. To do so they check if AD mutant mice, in which the dorsal nasal cavity is depleted of olfactory sensory neurons (some of them sensitive to TMT; Kowayakawa et al., 2007), show 2MT-induced hypothermia: there is a reduction of hypothermia in AD mice (Fig. 4B). However, depletion of *trpa1* in olfactory sensory neurons has no effect on 2MT-hypothermia. From these data, it seems that AD mice might have not just depletion of olfactory neurons in the dorsal olfactory epithelium but also altered trigeminal innervation of the nasal cavity.

Response: The 2MT-induced hypothermia is suppressed in ΔD mice. This can be explained by the hypothesis that the trigeminal pathway is affected in ΔD mice, as Reviewer#3 pointed out, or can be explained by the hypothesis that signaling by the olfactory receptors itself may impact. Our and other groups reported that the fear-related behaviors induced by 2MT, SBT, and TMT are regulated by a specific olfactory receptor and CNGA2 channels expressed in the olfactory neurons (references 6-8). Thus, it is possible that tFO-induced physiological responses is also regulated by olfactory receptors, in addition to *Trpa1* in the trigeminal and vagus nerves. To demonstrate this possibility more clearly, we have added a new experiment analyzing the 2MT-induced hypothermia in the ΔD (*cng*) mice, in which *Cnga2* is deleted only in the dorsal OSNs, in the revised manuscript. CNGA2 channel is specifically expressed in the OSNs among sensory nerves. In ΔD (*cng*) mice, axonal projections by OSNs and the formation of glomeruli on the olfactory bulb is normal; thus, it is expected that there is no direct effect on the trigeminal pathway. We observed suppression of 2MT-induced hypothermia also in the ΔD (*cng*) mice. Thus, it is likely that signaling by olfactory receptors, which require CNGA2 channels, is involved in regulating 2MT-induced hypothermia. We have added the results of the ΔD (*cng*) analysis in the revised manuscript (lines 206-210, Figure S4).

In summary, altogether the findings of the authors in such an impressive array of complex experiments suggests that trigeminal rather than olfactory innervation is responsible of the observed effects of outer exposure to tFOs in a semi-closed cage. Therefore, using the term odorants to name these substances in the current context is confusing and misleading for the reader. I would change the title and the rest of the text to clarify this issue.

Response: As Reviewer #3 pointed out, our results indicate that TRPA1 regulates the physiological responses induced by the thiazoline-related molecules. Fear-related behaviors induced by 2-methyl-2-thiazoline (2MT) and 2,4,5-trimethyl-3-thiazoline (TMT) are also reportedly regulated by TRPA1 in the trigeminal nerve (reference 9).

In general, molecules that induce physiological responses via TRPA1 are called “chemesthetic agents,” not odorant molecules. Therefore, 2MT and TMT should be referred to as chemesthetic agents rather than odorant molecules. On the other hand, TMT is a component of fox feces and has been used in several studies as an odorant molecule, along with 2-sec-butyl-2-thiazoline (SBT), an alarm pheromone secreted by mice. Furthermore, it has been reported that CNGA2 channels and a particular type of odorant receptor in sensory neurons in the olfactory epithelium’s dorsal zone are involved in the regulation of TMT- and SBT-induced fear-related

behaviors (references 6-8). Several of the thiazoline-related odorants, including 2MT used in this study, are used as food flavoring agents. Therefore, thiazoline-related molecules and other heterocyclic compounds simultaneously have two roles: 1) as odorant molecules that induce behavioral responses by activating odorant receptors, and 2) as chemesthetic agents that induce behavioral/ physiological responses through TRPA1.

It was recently reported that fear conditioning in the presence of 2MT induced a long-lasting fear-related physiological response in humans ¹. This effect could be regulated by odorant receptors or by *Trpa1*, although their contributions are unclear at present.

In light of these circumstances, we believe that it is not appropriate to refer to thiazoline-related molecules as odorant molecules, as Reviewer #3 pointed out. To address this, we have 1) changed the term “odorant” to “compound” throughout the revised manuscript to avoid singling out a particular sensory system; 2) used the name of a specific compound, such as 2MT; or 3) used the term tFO, a generic term for several thiazoline compounds. We clearly stated that “tFO” is a thiazoline-related innate fear-eliciting compound (lines 34 and 60).

REPRODUCIBILITY and STATISTICS

One of the issues that worries current science panorama is the lack of reproducibility. This is one of the reasons why methodology must be described in high detail. This is also one of the reasons why I don't like the use of lots of supplementary material, very often with few details on procedure.

Figure 2A shows that 2MT induces an increase in oxygen consumption in *trpa1*^{-/-} mice, but a decrease in *trpa1*^{+/-} ones. Consequently, survival in severe hypoxic conditions (4% oxygen) is very short in *trpa1*^{-/-} as compared to *trpa1*^{+/-} animals (Figure 2B). Figure 2D shows an additional experiment in which saline and acetaminophen (APAP) are administered to *trpa1*^{-/-} and *trpa1*^{+/-} mice and their survival in hypoxic conditions tested. APAP, a ligand of TRPA1, increases survival in control but not *trpa1*^{-/-} animals.

There is a lack of methodological details in the description of the experiment (in general in the experiments). Is 2MT exposure performed like in the experiment reported in Fig. 1 (soaking filter paper and putting 4 papers in a semi-closed cage)? Or like in Figures 3 and 7 (i.p injection)?

Response: As Reviewer#3 pointed out, the experimental details were lacking in the initial manuscript. For Figure 2A, two pieces of filter paper dropped with 271 umol of 2MT were presented in a metabolic chamber. For Figure 2B, a filter paper dropped with 271 umol of 2MT was presented for 10 min in an open cage. The experimental details are described in the revised manuscript (lines 478-482 and 485-488).

Statistics is also an important issue here. According to the figure legend, Student's t-tests are used. Many of the experiments reported require a two-way ANOVA, with genotype as one of the variables and treatment as the second one, instead of simple t tests (Fig. 1; Fig. 2D; Fig. 4; Fig. 5C and 5D). In other cases (e.g. Fig. 7A or 5B) and ANOVA followed by Dunnett's post-hoc (comparing each treatment with control), instead of simple (iterative) Student's t tests, or Bonferroni if comparisons between different treatments are intended. Experiment reported in figure 7E probably requires a Kaplan-Meier survival test.

There is no mention of normality and homoscedasticity requirements in any figure (required for t tests), and it is hard to believe that all the data fulfil these requirements in all experiments. In fact, there is subsection on statistics (just a simple paragraph) in the section of Methods.

Response: All the data were subjected to normality tests, and appropriate statistical calculations were performed according to these calculations' results. The specific statistics used are described in the "Statistics and reproducibility" section (lines 648-754) and each figure legend.

FIGURE 3 and brain activation by IP-injected tFOs

Figure 3 shows morphological details of the innervation of vagal and trigeminal centres by *trpa1*-expressing cells, and also of activity-related markers (p-ERK and *c-fos* mRNA) in response to IP injection of 4E2MT. Immunoreactivity for p-ERK and *c-fos* mRNA should be compared between 4E2MT- and saline-exposed animals. Otherwise we ignore if this labelling is due to 4E2MT or it is constitutive. Although the involvement of the vagal/trigeminal pathways on the biological effects of these drugs is likely, there is no clear demonstration of it in the results of the manuscript as currently reported.

Response: In response to this comment, we conducted a new experiment wherein we analyzed the expressions of *c-fos* mRNA and pERK in the saline-exposed/control condition. *C-fos* mRNA expression in the NTS and Sp5 in saline conditions is shown in revised Figure 3E. Representative images of immunohistochemical staining with pERK in the saline condition are shown in revised Figure 3G, and quantification of pERK positive cells for saline administration are compared to those for 4E2MT administration in Figure 3L.

BIOPROTECTIVE EFFECTS OF tFOs

I really don't grasp the use of the term "bioprotective" in the context of the experiments of the manuscript. Concerning anti-inflammatory actions of some tFOs, it is reported to reduce TNF-alpha in response to LPS injection. This is, per se, no bioprotective, as inflammatory responses are bioprotective or adaptive in some circumstances. Concerning the hypothermia induced by tFOs, it is bioprotective if there is fever or hyperthermia. Finally, survival under severe hypoxia is enhanced, and therefore this is a bioprotective effect in strange circumstances in which there is hypoxia. For instance, if there is stroke there cell survival under local hypoxia would arguably be bioprotective. But the trigeminal/vagal-mediated effects of tFOs are not possibly local.

Response: As Reviewer #3 pointed out, the physiological responses induced by tFOs are not always bioprotective. Therefore, in the revised manuscript, the term "bioprotective" has been deleted and replaced with "hypothermic and anti-hypoxic," which were effects measured in the experiments.

I can't understand the relationship of H2S-inhibition of mitochondrial respiration and the affair of tFOs. The paragraph of the Introduction on that issue is difficult to understand and very confusing: "Low concentration of H2S is thought to partially inhibit mitochondrial respiratory activity [22], thereby enhancing the ability to survive in a hypoxic environment". I cannot understand this statement that looks, at least, counter-intuitive.

Response: As Reviwer#3 pointed out, “low concentration of H₂S inhibit mitochondrial respiratory chain to enhance survivability in a hypoxic environment” is counterintuitive. However, it is well known that low concentration of H₂S enhance survivability in a hypoxic environment. To clarify this point, we corrected the placement of the citation in the revised manuscript. We also added a new reference on H₂S and hypoxia resistance (line 129).

CONCLUSION

The manuscript by Matsuo and collaborators reports a series of interesting experiments on a putatively promising issue (ligands of trpa1 receptors, and their biological effects). But it has important methodological problems related to statistics and experimental design (c-fos and p-ERK). In addition, there are also conceptual issues that need the attention of the authors: the repeated use of “odorant” to refer to trpa1-ligands, when their results indicate that they likely are vagal/trigeminal stimuli rather than olfactory ones. And the use of the expression “bioprotective effects” to refer to simple anti-inflammatory and hypothermic effects apparently mediated by (according to authors results) the effect of these drugs on peripheral fibres of the somatosensory system. The relationship between antipredatory (fear) reactions and this stimulation of vagal/trigeminal fibres is very unlikely, so that speaking of tFOs and mediating survival mechanisms in life-threatening situations seems not appropriate.

Response: Thank you once again for your consideration and careful review of our work. We hope that our responses here and the most recent version of our manuscript clear up the main issues you indicated in your feedback.

Reference

- 1 Taylor, J. E. *et al.* An Evolutionarily Threat-Relevant Odor Strengthens Human Fear Memory. *Front Neurosci* **14**, 255, doi:10.3389/fnins.2020.00255 (2020).

Reviewers' Comments:

Reviewer #2:

Remarks to the Author:

I believe the authors generally did a good job addressing my concerns.

There are some remaining questions that perhaps could be clarified without new experiments:

- can the differences in gene expression btw CNA and 2MT fully explain the differences in behavior?

Could it be that CNA activated a different population of Trpa1+ TG and VG neurons than 2MT? Perhaps a population that does not project to SP5 or NTS?

- is the 4E2MT induced c-fos expression in SP5 mediated by VG fibers?

- I don't think it is made clear in the abstract that, although OB does seem to mediate hypothermia, it does not do so via Trpa1.

There are some small typos and omissions (for example in Fig 3, I couldn't find an explanation for what the double arrow means). Also I think the order of the experiments presented is not always intuitive (for example, I would present the chemogenetic results after the description of TG and VG projections to the brainstem nuclei).

Line 292 - 'hypothermia' is written with a different font type.

I found it difficult to read the word document where all changes were tracked. On the other hand, in the pdf, not all changes were highlighted.

Reviewer #3:

Remarks to the Author:

The authors have coped with my suggestions and criticisms to my (virtually) complete satisfaction. I just have a few comments on specific points of the manuscript.

1. Description of the procedure to expose mice to volatile tFOs.

According to the description in Materials and Methods section of the revised manuscript, mice are exposed to 271 μmol of tFOs (2MT). The molecular weight of 2MT is 101.17g/mol (PubChem). This means that 271 μmol of 2MT equals to $271 \cdot 106 \cdot 101.17\text{g}$, 27.4 mg of the compound. Since the density of 2MT is 1.067 g/mL (data from Sigma-Aldrich), 271 μmol of 2MT (27.4 mg) correspond to $27.4/1.067 = 25.7 \mu\text{L}$ of pure 2MT. I need to do these calculations because otherwise it is not easy to understand exactly how experiments were performed and repeatability is compromised. For instance, in a very influential paper by the same group published some 13 years ago and cited in the manuscript (Kobayakawa, K. et al. Innate versus learned odour processing in the mouse olfactory bulb. Nature 450, 503-508, doi:10.1038/nature06281; 2007), the authors state that they expose animals to 20 μL of a 15,7M solution of TMT. With a molecular weight of 129.22 g/mol and a density of 1.013 g/mL, the molarity of pure TMT is of about 7,839M. This kind of mistakes are a headache for the reader, especially if he/she intends to reproduce and experiment. Therefore, I ask the authors to make sure that their calculations are correct and, maybe, to specify the volume of pure 2MT with which filter paper was soaked in the experiments (25.7 μL according to my calculations; is this correct?).

In contrast, description of the doses used to administer tFOs intraperitoneally is based on the volume of a 1% solution in saline (lines 439-440): "Administration of the compound was performed by intraperitoneally injecting 100 μL of 1% solution in saline". It would be nice to know what range of doses (mg/kg of body weight; even better, mol/kg of body weight) represents this volume/concentration injection. In a way, tFOs compounds are used here as pharmacological agonists of trpa1 receptors and, therefore, tFOs i.p. administration should be described as if it were drug administration, given a dose easy to extrapolate to other similar drugs. As I already suggested in my

previous report, given the low solubility of tFOs in water, maybe a 1% solution requires previously dissolving the compound in DMSO or absolute ethanol. Is this the case? If so, please specify it.

2. Figure 1: Lines 118-119 of the text

"Interestingly, hypothermia and bradycardia induced by 2MT tended to be greater in *Trpa1*^{-/-} mice than those in control mice." In Fig. 1A and 1B it seems just the opposite. *Trpa1*^{-/-} (red line) keeps about 37-38°C during 2MT exposure, whereas *trp1*^{+/-} (black line) drops to 35-36°C. A similar profile is observed in Fig. 1E, with heart rate dropping briskly from 700-800 bpm to 500 bpm in *trp1*^{+/-}, but being unaffected in *trp1*^{-/-} mice. I'm not sure if I'm right or I'm misunderstanding the text or the figure. I simply ask the authors to pay attention to this point.

3. Figure 2, and the corresponding part of Results section. The experiments of survival under strongly hypoxic conditions (4% O₂) should be probably analysed using a different statistical test. Apparently, many animals, especially in the group of APAP-treated mice, survive for the whole test (which seems to be 30 min, that is, 1800 s). Data are marginated, e.g. animals surviving for more than 1800s were assigned 1800s. Therefore, data are very unlikely to follow a normal distribution, when 3-6 animals have the same score (1800s, Fig. 2B, 2C and 2D) and, therefore, simple t tests or ANOVAs are not appropriate. That's probably why authors use a non-parametric Mann Whitney test, which seems not inappropriate. However, I still think that Kaplan Meier survival analysis, which are designed specifically for this kind of experiments, would be the best choice. In addition, I don't understand why authors use systematically (here and in many other experiments) one-tailed comparisons, instead of two-tailed ones. I would ask the authors to explain why in the new section on Statistics and Reproducibility in the section of Methods (pages 28-32). The authors should be complimented for adding this very informative section, which unfortunately, is very uncommon in this kind of papers.

I also find very clarifying the inclusion of new experiments of *cfos* expression using control (saline) animals together with tFOs-administered animals. I thank the authors for this addition, not very common in this kind of papers.

In the present conditions, I find that the manuscript is greatly improved and, to my view, it is now suitable to be published in a high-impact journal such as Nature Communications.

Congratulations for such a great work.

Yours

Fernando Martinez-Garcia

Point-by-point Response to Reviewers' Comments

Reviewer #2 (Remarks to the Author):

I believe the authors generally did a good job addressing my concerns.

There are some remaining questions that perhaps could be clarified without new experiments:

Thank you for your detailed critiques of our manuscript. We believe the reviewer's comments helped us improve the manuscript. In the revised manuscript, we have added two new experimental results, which we hope will allow us to answer in more detail the questions that the Reviewer #2 pointed out and that many readers will probably feel the same way about. We have made every effort to address the issues raised and to respond to all comments. The revisions are highlighted in the revised manuscript. Please, find a next detailed, point-by-point response to the reviewer's comments.

- can the differences in gene expression btw CNA and 2MT fully explain the differences in behavior? Could it be that CNA activated a different population of *Trpa1*⁺ TG and VG neurons than 2MT? Perhaps a population that does not project to SP5 or NTS?

Response: Thank you very much for thoughtful comment on differential effects of CNA and tFOs. As Reviewer #2 pointed out, it is unclear whether the difference in gene expression in the TG neurons between CNA and 2MT explains the differential responses between these compounds. We have added sentences to clarify this point (lines 348-352).

In addition to this, we have added two new experimental data that verify multiple possibilities that are currently conceivable. As reviewer#2 pointed out, different populations of *Trpa1*⁺ neurons may be activated by CNA and tFOs, and they may have different axonal projections in the brain. We thought it is important to verify this possibility and state the result in the manuscript. To do so, we have newly analyzed the calcium imaging of *Trpa1*⁺ TG/VG cells in the revised Supplementary Fig. 6. The responses for CNA and 4E2MT were simultaneously analyzed in 49 *Trpa1*⁺ VG cells; We found that almost all *Trpa1*⁺ VG neurons (48 out of 49 neurons) were more strongly activated by CNA than by 4E2MT, and the remaining one neuron responded almost equally to CNA and 4E2MT. We did not find any neurons which were activated by 4E2MT alone. Although we cannot exclude the possibility that a small population of *Trpa1*⁺ cells respond to tFOs, but not to CNA, our results do not support the model that specialized population of *Trpa1*⁺ neurons respond to tFOs but not to CNA. Alternatively, Sp5/NTS *c-fos* expression may be induced by non-TG/VG *Trpa1*⁺ sensory inputs

which are relevant to tFOs stimulation but not to CNA stimulation. To verify this possibility, we newly analyzed *c-fos* expression in the unilateral TG ablated mice in response to 2MT presentation. In the unilateral TG ablated mice, *c-fos* expression in the Sp5 in the lesioned side in response to 2MT presentation was suppressed compared to that in the contralateral side, suggesting that *c-fos* expression in the Sp5 is induced by ipsilateral TG projection neurons (Supplementary Fig. 7). We have added these statements to the revised manuscript (lines 307-324) and Supplementary Figs. 6, 7.

- is the 4E2MT induced *c-fos* expression in SP5 mediated by VG fibers?

Response: In GCaMP6 imaging, 4E2MT activated only *Trpa1*⁺ VG neurons but not *Trpa1*⁺ TG neurons (Fig 7c, d). Thus, as the reviewer pointed out, it is possible that *c-fos* expression in the Sp5/NTS induced by 4E2MT is dependent on the activation of *Trpa1*⁺ VG neurons. However, we speculate that the situation is a little bit more complicated. 5MT did not increase GCaMP6 signals in most *Trpa1*⁺ TG/VG neurons (Fig 7c, d), but it induced *c-fos* expression in the Sp5/NTS (Figure 8). Although 4E2MT stimulation did not induce calcium influx in the isolated *Trpa1*⁺ TG neurons, it induced expressions of immediate early genes in the TG (Figure 9). Thus, we speculate that GCaMP6 imaging using isolated TG/VG neurons may not reflect the *in vivo* response of these cells. In the current situation, it is difficult to explain the causality between GCaMP6 signaling in the *Trpa1*⁺ TG/VG neurons and *c-fos* expression in the Sp5/NTS. Accordingly, the limitation of GCaMP6 imaging using isolated TG/VG neurons are mentioned in the revised manuscript (lines 324-330).

- I don't think it is made clear in the abstract that, although OB does seem to mediate hypothermia, it does not do so via *Trpa1*.

Response: Following the reviewer's suggestion, we have added the sentence "TFO-induced hypothermia involves the *Trpa1*-mediated trigeminal/vagal pathways and non-*Trpa1* olfactory pathway." in the revised abstract (lines 37-38).

There are some small typos and omissions (for example in Fig 3, I couldn't find an explanation for what the double arrow means).

Response: We would like to thank the reviewer for pointing this out. We have added an explanation for double arrow in the legend to the revised Fig 4 (original Fig.3) in the revised manuscript (line1151-1152).

Also I think the order of the experiments presented is not always intuitive (for example, I would present the chemogenetic results after the description of TG and VG projections to the brainstem nuclei).

Response: We would like to thank the reviewer for the advice. Following the reviewer's suggestion, we have split the original Fig. 4 into two separate figures (revised Figs. 3 and 5) and changed the flow of manuscript to present (1) the contribution of three sensory system (lines 148-188; revised Fig. 3), (2) projection of *Trpa1*⁺ TG/VG neurons in the Sp5/NTS (lines 190-235; revised Fig. 4), and (3) chemogenetic results of *Trpa1*⁺ neurons projecting to Sp5/NTS (lines 237-258; revised Fig. 5) in the revised manuscript. We believe that our changes would help the readers understand the flow of our manuscript.

Line 292 - 'hypothermia' is written with a different font type.

Response: We would like to apologize to the reviewer for our mistake. Please note that we have corrected the font type in the revised manuscript (line 294).

I found it difficult to read the word document where all changes were tracked. On the other hand, in the pdf, not all changes were highlighted.

Response: We apologize for the difficulty to read the revised manuscript. Please note that we have highlighted all the changes made in the revised manuscript.

Reviewer #3 (Remarks to the Author):

The authors have coped with my suggestions and criticisms to my (virtually) complete satisfaction. I just have a few comments on specific points of the manuscript.

Response: Thank you for pointing out our insufficient explanation of experimental methods and statistical processing in our paper. By addressing the reviewer's comments, the manuscript were clearly improved. We have made every effort to address the issues raised and to respond to all comments. The revisions are highlighted in the revised manuscript. Please, refer to our detailed, point-by-point response to the reviewer's comments.

1. Description of the procedure to expose mice to volatile tFOs.

According to the description in Materials and Methods section of the revised manuscript, mice are exposed to 271 μmol of tFOs (2MT). The molecular weight of 2MT is 101.17g/mol (PubChem). This means that 271 μmol of 2MT equals to $271 \cdot 106 \cdot 101.17\text{g}$, 27.4 mg of the compound. Since the density of 2MT is 1.067 g/mL (data from Sigma-Aldrich), 271 μmol of 2MT (27.4 mg) correspond to $27.4/1.067 = 25.7 \mu\text{L}$ of pure 2MT. I need to do these calculations because otherwise it is not easy to understand exactly how experiments were performed and repeatability is compromised. For instance, in a very influential paper by the same group published some 13 years ago and cited in the manuscript (Kobayakawa, K. et al. Innate versus learned odour processing in the mouse olfactory bulb. Nature 450, 503-508, doi:10.1038/nature06281; 2007), the authors state that they expose animals to 20 μL of a 15,7M solution of TMT. With a molecular weight of 129.22 g/mol and a density of 1.013 g/mL, the molarity of pure TMT is of about 7,839M. This kind of mistakes are a headache for the reader, especially if he/she intends to reproduce and experiment. Therefore, I ask the authors to make sure that their calculations are correct and, maybe, to specify the volume of pure 2MT with which filter paper was soaked in the experiments (25.7 μL according to my calculations; is this correct?).

Response: As the reviewer pointed out, it is important to specify the method for the reproducibility and readability for readers. As calculated by the reviewer, we used 25.7 μL of 2MT to soak the filter paper. We have specified the volume of 2MT in the revised method (lines 519, 525, 560).

In contrast, description of the doses used to administer tFOs intraperitoneally is based on the

volume of a 1% solution in saline (lines 439-440): “Administration of the compound was performed by intraperitoneally injecting 100 µl of 1% solution in saline”. It would be nice to know what range of doses (mg/kg of body weight; even better, mol/kg of body weight) represents this volume/concentration injection. In a way, tFOs compounds are used here as pharmacological agonists of trpa1 receptors and, therefore, tFOs i.p. administration should be described as if it were drug administration, given a dose easy to extrapolate to other similar drugs. As I already suggested in my previous report, given the low solubility of tFOs in water, maybe a 1% solution requires previously dissolving the compound in DMSO or absolute ethanol. Is this the case? If so, please specify it.

Response: The tFO used for i.p. administration was approximately 40 mg/kg. We have provided this information in the revised method (lines 472, 520, 551, 600, 711). 2MT and TMO can be dissolved in saline at 1%. For the other compounds, because of their low solubility in saline, 1% of the compound was added to saline and stirred vigorously by vortex to ensure that the compound was dispersed in the saline just prior to IP administration. By this operation, 1 µL of the compound was administered into the abdominal cavity. We added this information in the revised method (lines 473-476).

2. Figure 1: Lines 118-119 of the text

“Interestingly, hypothermia and bradycardia induced by 2MT tended to be greater in *Trpa1*^{-/-} mice than those in control mice.” In Fig. 1A and 1B it seems just the opposite. *Trpa1*^{-/-} (red line) keeps about 37-38°C during 2MT exposure, whereas *trp1*^{+/-} (black line) drops to 35-36°C. A similar profile is observed in Fig. 1E, with heart rate dropping briskly from 700-800 bpm to 500 bpm in *trp1*^{+/-}, but being unaffected in *trp1*^{-/-} mice. I’m not sure if I’m right or I’m misunderstanding the text or the figure. I simply ask the authors to pay attention to this point.

Response: We would like to thank the reviewer for pointing this out. Please note that we have revised the corresponding part as “Interestingly, hypothermia and bradycardia induced by restraint in a tight space tended to be greater in *Trpa1*^{-/-} mice than those in control mice” in the revised manuscript (line 117-119).

3. Figure 2, and the corresponding part of Results section. The experiments of survival under strongly hypoxic conditions (4% O₂) should be probably analysed using a different statistical test. Apparently, many animals, especially in the group of APAP-treated mice, survive for the whole test (which seems to be 30 min, that is, 1800 s). Data are marginated, e.g. animals surviving for more than 1800s were assigned 1800s. Therefore, data are very unlikely to follow

a normal distribution, when 3-6 animals have the same score (1800s, Fig. 2B, 2C and 2D) and, therefore, simple t tests or ANOVAs are not appropriate. That's probably why authors use a non-parametric Mann Whitney test, which seems not inappropriate. However, I still think that Kaplan Meier survival analysis, which are designed specifically for this kind of experiments, would be the best choice.

Response: Following the reviewer's suggestion, we have reanalyzed the result in Fig. 2b-d using the log-rank test. We have provided the statistics used and reanalyzed *p* values in the revised manuscript (lines 742, 743, 746, 1107-1115).

In addition, I don't understand why authors use systematically (here and in many other experiments) one-tailed comparisons, instead of two-tailed ones. I would ask the authors to explain why in the new section on Statistics and Reproducibility in the section of Methods (pages 28-32). The authors should be complimented for adding this very informative section, which unfortunately, is very uncommon in this kind of papers.

Response: In response to the reviewer's comment, we have mentioned the reason for using one-tailed comparison in the statistics section (lines 777-780, 782-783, 792-794, 797-798, 817-819, 821-822, 826-831, 836-837, 846-847, 853-854).

I also find very clarifying the inclusion of new experiments of cfos expression using control (saline) animals together with tFOs-administered animals. I thank the authors for this addition, not very common in this kind of papers.

In the present conditions, I find that the manuscript is greatly improved and, to my view, it is now suitable to be published in a high-impact journal such as Nature Communications.

Congratulations for such a great work.

Yours

Fernando Martinez-Garcia

Response: We would like to thank the reviewer for the positive evaluation and his kind comments. We hope that the revisions meet the reviewer's requirements.